# Dynamic self-stabilization in the electronic and nanomechanical properties of an organic polymer semiconductor

Illia Dobryden [1,2,8], Vladimir V. Korolkov [3,8✉], Vincent Lemaur[4], Matthew Waldrip [5], Hio-Ieng Un[6], Dimitrios Simatos [6], Leszek J. Spalek[6], Oana D. Jurchescu [5], Yoann Olivier[7], Per M. Claesson[1] & Deepak Venkateshvaran [6✉]

The field of organic electronics has profited from the discovery of new conjugated semi-conducting polymers that have molecular backbones which exhibit resilience to conformational fluctuations, accompanied by charge carrier mobilities that routinely cross the 1 cm$^2$/Vs benchmark. One such polymer is indacenodithiophene-co-benzothiadiazole. Previously understood to be lacking in microstructural order, we show here direct evidence of nanosized domains of high order in its thin films. We also demonstrate that its device-based high-performance electrical and thermoelectric properties are not intrinsic but undergo rapid stabilization following a burst of ambient air exposure. The polymer's nanomechanical properties equilibrate on longer timescales owing to an orthogonal mechanism; the gradual sweating-out of residual low molecular weight solvent molecules from its surface. We snapshot the quasistatic temporal evolution of the electrical, thermoelectric and nano-mechanical properties of this prototypical organic semiconductor and investigate the sub-tleties which play on competing timescales. Our study documents the untold and often overlooked story of a polymer device's dynamic evolution toward stability.

[1] Division of Surface and Corrosion Science, Department of Chemistry, School of Engineering Sciences in Chemistry, Biotechnology and Health, KTH Royal Institute of Technology, Drottning Kristinas väg 51, SE-100 44 Stockholm, Sweden. [2] Experimental Physics, Division of Materials Science, Department of Engineering Sciences and Mathematics, Luleå University of Technology, SE-971 87 Luleå, Sweden. [3] Park Systems UK Limited, MediCity Nottingham, Thane Road, NG90 6BH Nottingham, UK. [4] Laboratory for Chemistry of Novel Materials, University of Mons, Place du Parc 20, B-7000 Mons, Belgium. [5] Department of Physics and Center for Functional Materials, Wake Forest University, Winston-Salem, NC 27109, USA. [6] Cavendish Laboratory, University of Cambridge, JJ Thomson Avenue, CB3 0HE Cambridge, UK. [7] Laboratory for Computational Modelling of Functional Materials, Namur Institute of Structured Matter, Université de Namur, Rue de Bruxelles, 61, B-5000 Namur, Belgium. [8] These authors contributed equally: Illia Dobryden, Vladimir V. Korolkov. ✉email: vladimir@parksystems.com; dv246@cam.ac.uk

Organic semiconducting polymers based on donor-acceptor motifs have risen to prominence in the field of organic electronics for their high charge carrier mobilities that approach and sometimes exceed $1 \, cm^2/Vs$, and for their ability to show ambipolarity by transporting both negatively charged electrons as well as positively charged holes[1–8]. A powerful subset in this family of donor-acceptor co-polymers is one in which a planar backbone constitutes the donor segment of the polymer's monomer unit to limit torsion within it[5,9]. Should the stiff donor unit be chemically engineered to bond with an acceptor unit via a single bond that supports resilience to conformational fluctuations, the polymer becomes efficient at transporting charge carriers along its backbone[10]. Such a sought-after molecular arrangement within a donor-acceptor based conjugated organic polymer is visible in the archetypical organic polymer indacenodithiophene-co-benzothiadiazole, also known as C16-IDTBT[9,11–14]. Its molecular design consistently guarantees charge carrier mobilities larger than $1 \, cm^2/Vs$ and has inspired a range of new organic semiconducting co-polymers based on stiff or fused backbones[15]. C16-IDTBT's nanoscale morphology contains nanoscale regions of both order and disorder, as conclusively shown in this work, with minute regions of high ordering on a scale of ∼15 nm within device-grade solution processed thin films. Low torsional disorder in C16-IDTBT and its influence on charge transport were previously investigated using two electronic device-based approaches[9,16]. First, field-effect transistors showed that the scaling between the drain current and the gate voltage in the saturation regime conformed to an ideal behavior described by the MOSFET equation, indicative of low energetic disorder. Second, a gate voltage modulated Seebeck coefficient measurement within the channel of the organic field-effect transistor showed that most carriers induced in the channel of the device contribute to entropy transport, and only a small fraction of these accumulated charges remain trapped in immobile electronic states. In this work, we spotlight a crucial yet missing piece in the puzzle of charge transport within C16-IDTBT based electronic devices by demonstrating how its device characteristics show rapid improvement toward ideality during the hours immediately after fabrication and upon ambient exposure. While significant strides have been made to stabilize trap-free hole transport in C16-IDTBT based electronic devices through the incorporation of molecular additives and solvent additives[11,12,17], we demonstrate here how such stabilization can be self-induced over time in the presence of ambient conditions alone. We expand our understanding of the multifunctional properties of C16-IDTBT by probing its nanomechanical properties at high-resolution and study their evolution in time. The nanomechanical properties of the polymer stabilize with time to display texture, although this stabilization does not depend on ambient air exposure. Our finding points to an orthogonal mechanism based on the gradual sweating out of residual low molecular weight additive solvent molecules from the polymer film. These residual solvent molecules have temporary, additive-like stabilizing effects; they can neutralize moisture-based traps, confirmed to exist in C16-IDTBT devices, through the formation of solvent-water azeotropes[17,18]. A complete electrical, thermoelectric and nanomechanical property characterization within thin films and devices fabricated from the archetypical polymer C16-IDTBT together with their time evolution is an important step toward its application in next-generation air-stable polymer semiconductor based electromechanical devices[19]. Our work emphasizes the prime importance of strictly connecting the measured properties of an organic multifunctional device together with the ambient conditions under which they are reported, since organic materials very often show a dynamic evolution. It also highlights the importance of reporting all details of device fabrication and characterization procedures to ensure better reproducibility between different labs.

## Results and discussion

**Direct evidence of nanoscale order in device-grade C16-IDTBT thin films**. In addition to its torsion resilient molecular backbone that manifests high mobilities, C16-IDTBT is understood to have a largely amorphous microstructure with no long-range order[9]. This understanding has been challenged recently through high resolution transmission electron microscopy (TEM) that showed ordering on ultra-small length scales of tens of nanometers[20]. The film preparation for such TEM measurements is nontrivial and necessitates a film float-off and transfer onto a TEM grid, a process that will intrinsically alter the mechanical properties of the film under investigation. To preserve the pristine organic film's large-area structural integrity, and to maintain consistency across the structural, electrical, thermoelectric and nanomechanical measurements, we carry out all our investigations on identical device-grade solution processed thin films of C16-IDTBT (see Supplementary Information Section 1). Figure 1(a) shows the chemical structure of C16-IDTBT containing a stiff backbone donor indacenodithiophene (IDT) unit co-polymerized with an acceptor benzothiadiazole (BT) unit. Figure 1(b) and (c) respectively show images of the height topography and its corresponding phase contrast scanned under ambient conditions using an atomic force microscope (AFM) over an area of 350 nm × 350 nm in a stabilized solution processed thin film of C16-IDTBT (see Methods Section for the higher eigen mode imaging technique used). At this resolution, the phase contrast image already begins to manifest tiny regions where aligned polymer filaments are observed. A high-resolution image of the phase contrast having a scale bar of 20 nm is shown in Fig. 1(d). Here, the real space map of the polymer surface clearly shows two regions (encircled in white) within 30 nm of each other. One has significant order, while the other shows unclear ordering. The long bright streaks seen in the region of high order are the molecular faces of individual polymer backbones. Six additional scanned regions of the film showing nanoscale order are summarized in the Supplementary Information (see Supplementary Information Section 2). Unlike the semiconducting polythiophenes whose polymer backbones are stacked at an angle to the substrate[21–23], C16-IDTBT has its molecular backbone face-down on the substrate with its dominant $\pi - \pi$ stacking direction perpendicular to the substrate[5]. A scan along the dotted white line in the ordered region of C16-IDTBT within the white circle of Fig. 1 (d) reveals a periodicity of 1.6 nm as shown in Fig. 1(e). This periodicity within the top surface of the thin film and on the nanoscale, does not confirm the findings of high-resolution scanning tunnelling microscopy measurements (STM) in dispersed molecules of C16-IDTBT which show sidechain interdigitation[6]. Rather, it confirms the recent investigation using TEM where the ordered phase of C16-IDTBT within spin coated thin films show a liquid crystalline-like character with no interdigitation of the alkyl side chains within the plane of the film. The polymer backbones are nevertheless aligned in parallel with their molecular faces looking up. The bright regions of the polymer backbones are assumed to be from the non-planar $sp^3$ bridging carbons in the IDT units[6]. Fig. 1(f) is a line scan across the encircled disordered region of Fig. 1(d) and shows the absence of such periodicity. The regions of nanoscale order shown for the first time using the AFM here provide, simultaneously, confirmation of the length scale over which order is observed using high resolution TEM[20], and of the parallel polymer backbone filament arrangement observed using high resolution STM[6]. The highlight, as should be reiterated, is that the 50 nm thick films

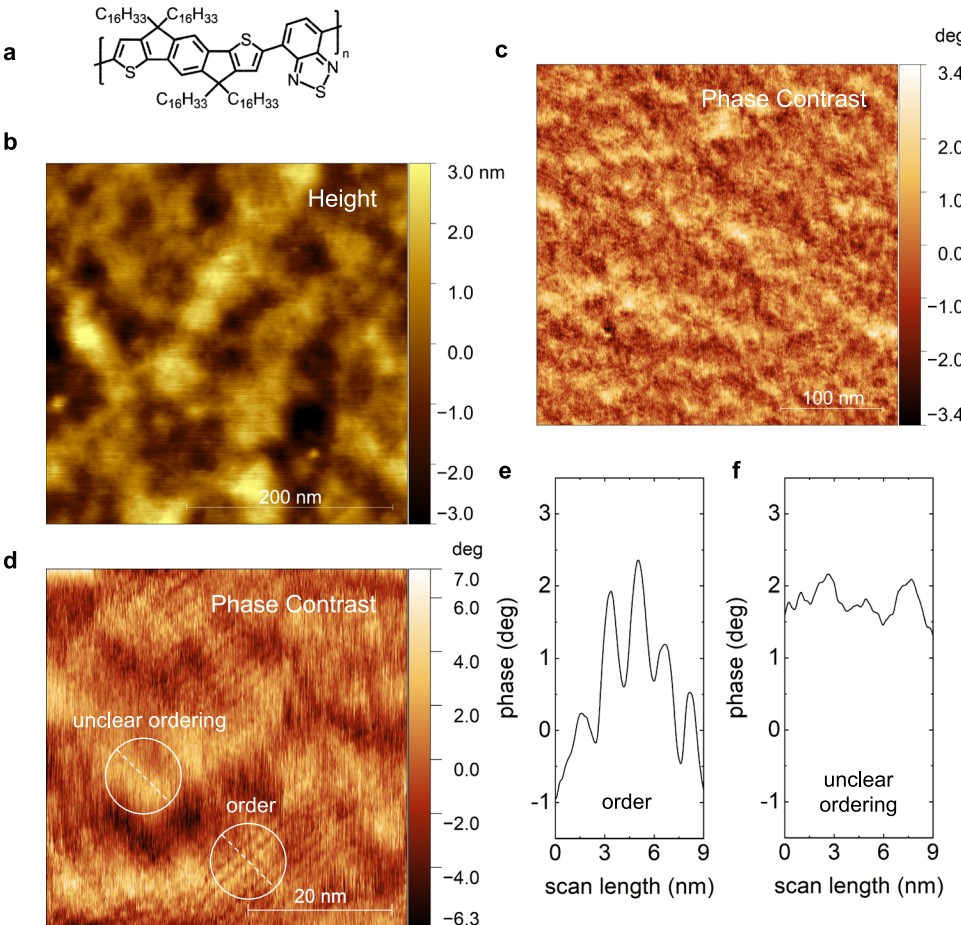

**Fig. 1 Nanostructure of C16-IDTBT films. a** Chemical structure of a C16-IDTBT monomer showing its fused donor molecular unit (IDT) and its acceptor molecular unit (BT). **b** Height topography of a C16-IDTBT spin-coated thin film. **c** Corresponding phase contrast image of the C16-IDTBT thin film on the scale of hundreds of nanometers. **d** Phase contrast image on the scale of tens of nanometers demonstrating regions of order 10–15 nm across and regions of unclear ordering within tens of nanometers from the ordered regions. **e** Line scan along the dashed straight lines of (**d**) in the ordered region. **f** Line scan along the dashed straight line of (**d**) in the region of unclear order. The distance between the parallel polymer backbone filaments of (**d**) is ~1.6 nm and is smaller than that expected for in-plane sidechain interdigitation.

investigated in Fig. 1 were spin-coated using the common parameters deployed for C16-IDTBT based organic transistors and require very elementary sample preparation.

**Temporal evolution of electrical and thermoelectric properties in C16-IDTBT based devices**. The electrical and thermoelectric properties of C16-IDTBT are studied through measurements of organic thin film transistor characteristics and gate voltage modulated Seebeck coefficient within the same device (see Supplementary Information Section 3, 4, and 5)[8,9,16,24–26]. Figure 2(a) shows the device schematic and measurement configuration of the transistor transfer and output characteristics measured in this work (see Methods Section for its concise fabrication). The Seebeck coefficient is measured by sending a current through the stripe heater, shown in Fig. 2(b), and by measuring the open circuit built-in thermal voltage between the source and the drain in the regime of high carrier accumulation. The ratio between the open circuit thermal voltage and the temperature difference across the device channel is the Seebeck coefficient. Figure 2(c)–(e) document the time evolution of the transistor transfer characteristics, transistor output characteristics and the gate voltage modulated Seebeck coefficient within the C16-IDTBT device upon exposure to ambient air. The time stamp on each snapshot refers to the cumulative time that the device spent in ambient air from the point of the first measurement immediately after fabrication.

Figure 2(c) and (d) show that the device 'heals' over time, from non-ideal to ideal transistor characteristics. Right after device fabrication, the on-current is relatively low. After a month of ambient exposure, it increases by two orders of magnitude in the saturation regime (even more in the linear regime), and the subthreshold swing reduces to ~800 mV/decade. This is the smallest value ever reported for this polymer. Existing literature report sub-threshold swings that span a few V/decade, all the way up to 5 V/decade in C16-IDTBT transistors at room temperature[5,6,11,13,14,27–30]. After a month of ambient air exposure, textbook-like output characteristics are also observed. The increased on-current, reduced subthreshold swing, positive turn-on voltage, and rapidly rising current in the low drain and low gate voltage regimes all indicate that the device becomes less resistive and less trap-limited in its charge transport, with improved charge extraction and injection over time. On the other hand, devices which were not exposed to ambient air and stored in a nitrogen glovebox over the same period continue to present current-voltage characteristics that are non-ideal. In addition, the healed devices partially return to showing non-ideal characteristics when left in vacuum over long durations as the adsorbed air is removed from the device. These phenomena reveal that ambient air is responsible for the improvement, presumably,

**a** transistor characteristics **b** Seebeck coefficient

**c** transistor transfer characteristics with time

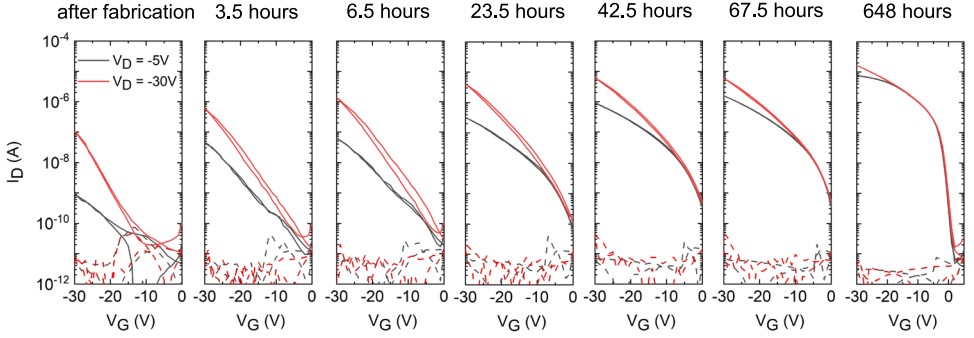

**d** transistor output characteristics with time

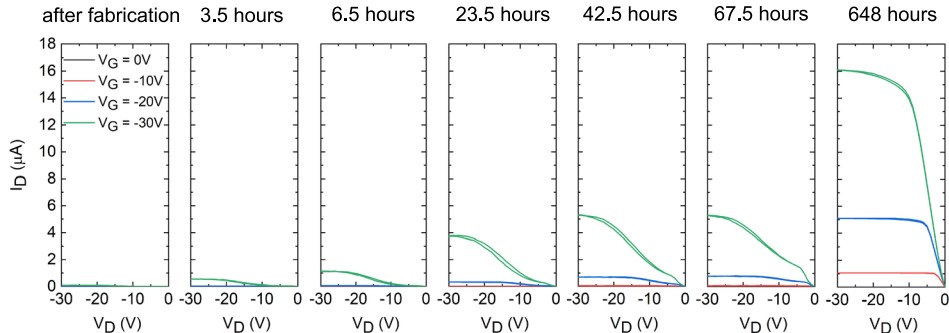

**e** gate voltage-modulated Seebeck coefficient with time

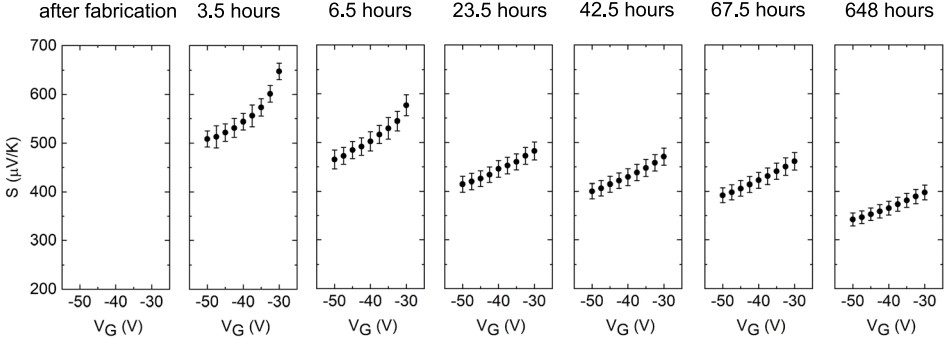

**Fig. 2 Time evolution of the electrical and thermoelectric properties of C16-IDTBT. a** Device configuration for the measurement of transistor transfer and output characteristics. **b** Device configuration for the measurement of the gate voltage modulated Seebeck coefficient. The gradient arrow indicates the direction of the positive temperature difference. **c** Transistor transfer characteristics in both the linear regime and the saturation regime over time. The dotted lines are the gate leakage currents in the dielectric. **d** Transistor output characteristics over time. **e** Gate voltage modulated Seebeck coefficient measured over time in ambient air, in which the error bars arise from the linear fit of the thermal voltage to the applied temperature differential (See Supplementary Information Section 5).

through a p-doping/electron-trapping mechanism involving oxygen[31–33]. In addition, the initial increase in the onset voltage, which is the signature of doping, followed by a decrease over time [Fig. 2(c)], suggests that two different processes occur during device aging. Indeed, the density of trap states (t-DOS) spectra confirm the presence of two regimes (see Supplementary Information Section 6)[18,27]. Molecular dynamics simulations on a C16-IDTBT amorphous morphology in the presence and absence of oxygen reveal no change in the donor-acceptor torsion distribution and so the charge transport properties along the polymer backbone are not affected by conformational effects due to the presence of oxygen (see Supplementary Information Section 7). Still, earlier reports in the literature have highlighted that oxygen can hybridize with the conjugated backbone of a polymer to form a weak charge transfer state and further capture an electron by converting to $O_2^{-}$[34–37]. C16-IDTBT is a hole-transporting material with the lowest occupied molecular orbital (LUMO) level of –3.6 eV[38], well-aligned with the reduction potential of oxygen without water and well beyond the reduction potential of oxygen with water; this very likely leads to a result that the minority carriers (electrons here) of C16-IDTBT get trapped by the electrochemical reaction of oxygen with or without water and, simultaneously, the hole carrier concentration increases[39–41]. Such an increase in hole carrier concentration can effectively fill the low-mobility electronic states and enable mobile carriers to access higher energy states with higher mobility[42]. This picture is consistent with our results showing that the saturation hole mobilities extracted from the high gate voltage regime improve to a high value approaching 1 cm²/Vs within the first 24 h of exposure, as discussed later. The resulting trap-filling effect also decreases bulk trapping (bulk resistance), in staggered transistors (i.e., Top Gate Bottom Contact that was used here), and thus decreases the contact resistance that consists of the interface resistance and the bulk resistance. A recent study reported that p-channel transistors can be improved by a spontaneous p-doping through well-aligned interface dipoles of a fluorinated dielectric[29], but this mechanism cannot entirely explain our time-dependent improvement because the polarization induced at the dielectric-semiconductor interface is expected to be a much faster process. The healing of the non-ideal behavior and the improvement in electrical properties of C16-IDTBT transistors that occur majorly within the first few hours of air exposure have gone unreported till date.

Doping induced by ambient exposure or controlled exposure to ppm quantities of ozone are known to cause a shift in the threshold voltage in organic transistors toward more positive values as reported in some polythiophenes[43]. Such a doping process is normally accompanied by degraded transistor performance, evidenced by high off-currents and by sub-threshold slopes that get shallower upon exposure. In our work, the observation that the threshold voltage initially becomes more positive does indeed indicate doping, but the observation that it eventually reverses direction within a month accompanying a steepening in the subthreshold slope is an indication that, at the very least, one other accompanying process takes place. We speculate that this accompanying process might be related to a structural reorganization within the film as the residual solvent contained within it sweats out over time. Evidence of the same is presented in the context of the nanomechanical property measurements later in this paper.

The Seebeck coefficient is intrinsically a measure of the entropy per unit charge in the device. To better understand how ambient air exposure influences the electronic states and thus charge transport in C16-IDTBT, we measured the Seebeck coefficient as functions of carrier concentration (regulated by gate voltage) and

ambient air exposure time. As mentioned above, the device is very resistive immediately after fabrication, making the Seebeck coefficient difficult to measure reliably; the measurement as a result starts from an exposure of 3.5 h, with applied gate voltage between –30 V and –50 V which is well beyond the turn-on voltage[16]. Figure 2(e) shows a reduction of the Seebeck coefficient as a function of exposure time. The reduction with time is maximum at low gate bias (from ca. 650 to 400 μV/K), as opposed to high gate bias (from ca. 500 to 350 μV/K). Since a reducing Seebeck coefficient is an indicator of an increasing number of holes that participate in energy and entropy transport, this shows that initially the device has more carriers in trap states which do not contribute to entropy transport at low carrier accumulation. It may also be a simultaneous indication that the injection and extraction of charge carriers improves over time. Figure 2(e) also reveals that the most rapid change in the value of the Seebeck coefficient occurs within the first 24 h of air exposure after which the change is gradual.

Figure 3 plots the salient features of the electrical transport and the Seebeck coefficient from Fig. 2. Figure 3(a) shows the region of the output curves within the first –1 V of the applied drain voltage and under a low gate voltage bias (low hole accumulation) of –10 V. A rapidly rising current upon air exposure is an indication of gradually improved contact properties (carrier injection and extraction) within the device. This presents indirect evidence for a reduction in the contact resistance upon ambient exposure. The ambient improvement of the contact resistance within the C16-IDTBT device is qualitatively equivalent to a mechanism that uses a molecular dopant for trap healing[44]. Figure 3(b) and (c) plot the output characteristics of the device under a large gate voltage bias of –30 V (high hole accumulation) and for drain voltages up to only –10 V. The device currents in this regime present kinks in the output characteristics that appear to vanish over time upon air exposure. On increasing the drain voltage from 0 V, the current rises sharply at low $V_D$ but sluggishly thereafter. It then shows a sudden and continuous increase at higher drain voltages (before eventually plateauing out at even larger $V_D$). The device measured after a month does not have such a non-ideality. These kinks are seen well beyond the region around 0 $V_D$, indicating that its origin is distinct from the issue related to contact resistance and injection. Some papers report that kinks in transistor output characteristics are typically a sign of the presence of ambipolar charge transport[45,46]. However, no supporting evidence can be found of there being mobile electrons in our devices in the regime of measurement. A plausible mechanism is thus a (multistage) non-equilibrium carrier trapping with field- and phonon-assisted de-trapping processes that involves trap states[47,48]. When the applied drain voltage is sufficiently high, the trapped carriers start to escape from traps, resulting in a sudden increase in the current. The suppression of the kink after a month of air exposure suggests that there are sufficient doping-induced carriers to fill up those traps so that there is no observable impact on the transport of injected carriers, consistent with our previous discussion. It is also in agreement with the reduction in the trap density in the intermediate energy regime (see Supplementary Information Section 6). To our knowledge, the device's transient non-idealities observed here after fabrication are qualitatively different from the non-idealities routinely studied in organic transistors in the past[49,50], but a detailed investigation on this kind of non-ideality is beyond the scope of the current work.

Figure 3(d) showcases the saturation mobility extracted within the C16-IDTBT device. The mobility was extracted using a few data points at the maximum $V_G$ of –30 V, and for $V_D$ = –30 V. In the linear-log plot of the mobility with time, the

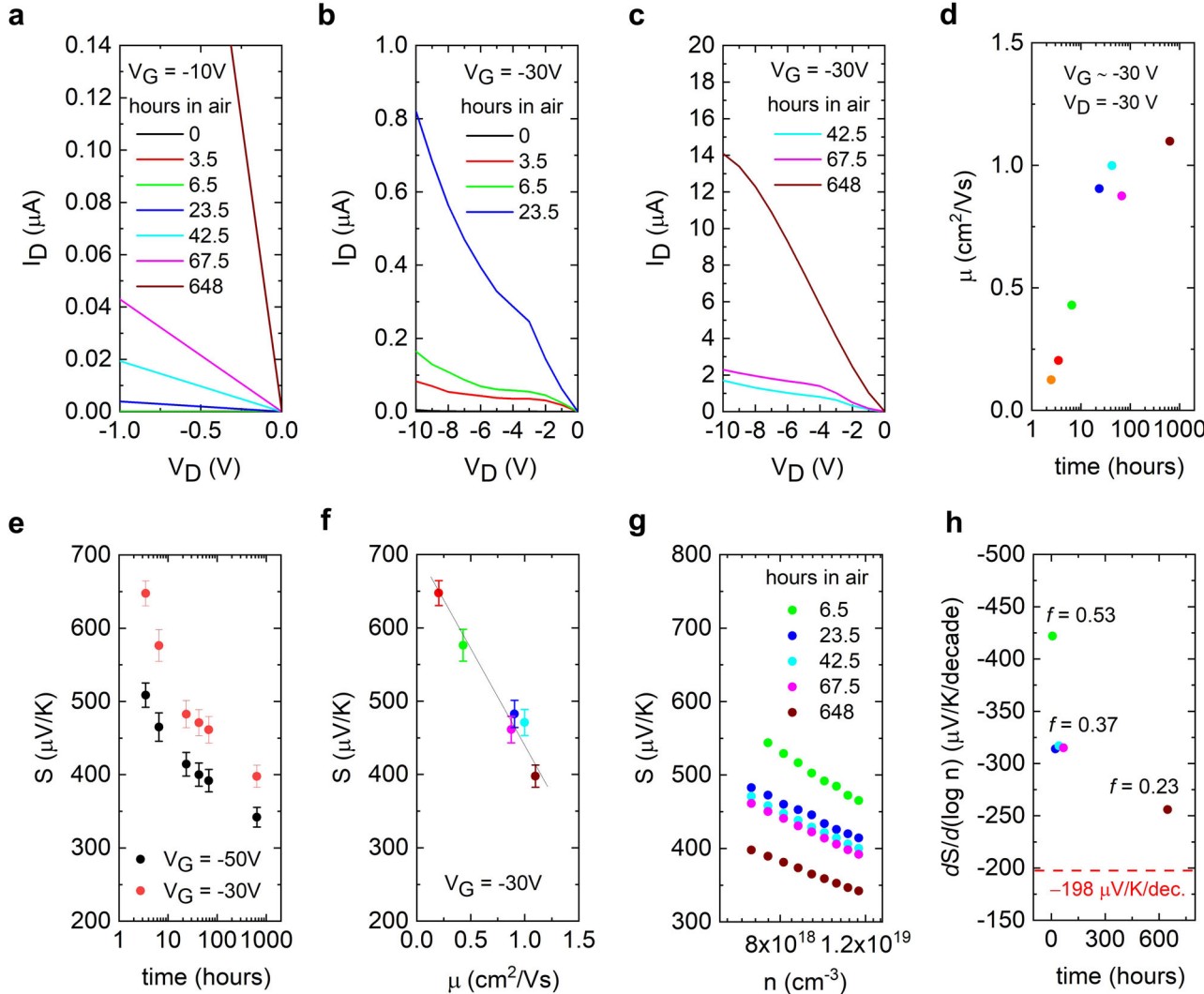

**Fig. 3 Parametric dependence of mobility and Seebeck coefficient on time. a** Output characteristics around $V_D = 0$ V for low gate voltage of $V_G = -10$ V.
**b** Output characteristics for intermediate drain voltages $V_D = 0$ V to $V_D = -10$ V, high gate voltage of $V_G = -30$ V, and for up to one day in ambient air.
**c** Output characteristics for intermediate drain voltages $V_D = 0$ V to $V_D = -10$ V, high gate voltage of $V_G = -30$ V, and for between 2 days and a month in ambient air. (**d**) Time evolution of the saturation mobility in the transistor. **e** Time evolution of the Seebeck coefficient in the transistor channel.
**f** Parametric dependence of saturation mobility and Seebeck coefficient on time over three temporal decades in hours (**g**) Seebeck coefficient vs. carrier density in the transistor. **h** Time evolution in the slope of the Seebeck coefficient vs. carrier density in the transistor channel.

hole mobility appears to rapidly increase within the first ten hours, but tails off within a day after achieving a mobility around 1 cm²/Vs. This is the hole mobility value that has been routinely reported in the literature, although higher values can be obtained upon aggressively reducing the contact resistance[30]. For durations beyond the first 10 h, the hole mobility appears to hover around this benchmark. Figure 3(e) plots the Seebeck coefficient as a function of time within the device for two gate voltages, $V_G = -30$ V and $V_G = -50$ V. In both cases, it is evident that the Seebeck coefficient shows two linear regimes on this linear-log plot. The first regime is within the first 10 h, while the second regime is beyond 10 h. The Seebeck coefficient appears to rapidly reduce in the first regime but slowly reduce in the second regime. The temporal regimes that govern the trajectory in the Seebeck coefficient coincide with the temporal regimes seen in the mobility evolution of Fig. 3(d). The linear trends in the plots of the mobility and Seebeck vs. time on a linear-log scale point at an exponential nature of change in these transport coefficients. Figure 3(f) is a plot of the Seebeck coefficient vs. the saturation mobility. The Seebeck coefficient

plotted here is for a gate bias of $V_G = -30$ V, to maintain consistency with the applied gate voltage of the saturation mobility against which it is plotted, i.e., $V_G \sim -30$ V and $V_D = -30$ V. From Fig. 3(f), irrespective of the two temporal regimes highlighted earlier, the Seebeck coefficient appears to be a linear function of the saturation mobility. In the absence of a known physical model that draws a linear dependence between these two transport coefficients in organic semiconductors within a non-degenerate hopping transport regime, the take home message is that there exists a parametric dependence of both the saturation mobility and the Seebeck coefficient on time. This parametric dependence is universal and applies over the investigated time scale from device fabrication all the way till the first one thousand hours in ambient air. Figure 3(g) plots the gate voltage modulated Seebeck coefficient as a function of calculated carrier density in the channel. The plot covers the measured regime of applied gate voltage between $V_G = -30$ V and $V_G = -50$ V, well beyond the device turn-on voltage. Both the magnitude of the Seebeck coefficient as well as its slope with carrier density on this linear-log plot reduce with time. The

reduction is once again logarithmic over three decades in time (in hours), with the measurements performed between 10 h and 100 h bunched together. The carrier density used in this plot was calculated using the capacitance of the dielectric and the applied gate voltage (See Supplementary Information Section 8). Within a thermoelectric transport model for C16-IDTBT developed earlier that assumes carrier transport in narrow bands, the slope of the Seebeck coefficient vs. channel-accumulated carrier density on a linear-log plot is an indirect pointer at the fraction of charge carriers that do not participate in entropy transport within the device. Should all the channel-accumulated charge carriers participate in transport of entropy without being trapped, the slope of the Seebeck coefficient vs. charge carrier density on a linear-log scale achieves its fundamental limit at $-k_B/e \times \ln(10) = -198\,\mu$V/K/decade. This limit is shown in Fig. 3(h) using a dotted red line. The slopes of the Seebeck coefficient with carrier density from Fig. 3(g) are plotted in Fig. 3(h) and demonstrates that the fraction of holes which participate in entropy transport within the device become progressively larger as the device characteristics improve upon ambient exposure. In the presence of induced carriers that do not contribute to entropy transport, the slope of the S–log($n$) plot is increased to $-k_B/e \times \ln(10)/(1-f)$, where $f$ is a qualitative measure of the trapped carriers. A lower $f$ parameter would mean more free carriers[9]. As a guideline, the $f$ parameter spotlight in our earlier work on the Seebeck coefficient in C16-IDTBT was $f = 0.3$[9]. This is because the measurements were performed within a few days of device fabrication. We show here that this value is improved even further to around $f = 0.2$ if the device is left in ambient conditions for durations that encompass the third decade in time (in hours). In the third temporal decade from fabrication, the electronic properties show improvement across the board, from electrical characteristics that include a small subthreshold slope, high hole currents and mobilities, and thermoelectric properties that report reduced Seebeck coefficients and larger fractions of carriers that contribute to entropy transport within the device. The fact that the saturation mobility and the Seebeck coefficient are linearly tied to each other would also imply that a completely healed C16-IDTBT device would fixate at a mobility of $\sim$1 cm$^2$/Vs and a Seebeck coefficient of $\sim$400 $\mu$V/K on Fig. 3(f).

The ideal slope of the Seebeck coefficient on a Jonker plot, i.e., $S - \log(n)$ or $S - \log(\sigma)$, is $-\frac{k_B}{e}\ln(10)$ within the non-degenerate transport regime where the carrier density $n$ is well below the total number of available states $N$. This is typically the case in our Cytop-gated organic field-effect transistors where the carrier density is around $10^{18}$–$10^{19}$ cm$^{-3}$. In systems such as ion-gel electrolyte gated organic transistors, the carrier density is typically an order of magnitude larger than it is in Cytop-gated organic field-effect transistors. At such high carrier densities which approach a 'near-degenerate' transport regime, the slope of the Seebeck coefficient is known to deviate and be shallower than $-\frac{k_B}{e}\ln(10)$. Within a transport model for polaronic hopping that has routinely been used to explain transport in multifunctional conductive oxide-based semiconductors, the magnitude of reduction in the Seebeck slope is known to depend sensitively on the dimensionless quantity $ne\alpha$. Here, $\alpha$ is the proportionality constant between the carrier mobility and the conductivity in a high conductivity regime, $e$ is the electronic charge and $n$ is the carrier density. In high conductivity polaronic systems, this reduced Seebeck slope becomes $-\frac{k_B}{e}\ln(10) \times (1 - ne\alpha)$, and has been observed in several multifunctional conductive oxide-based semiconductors, as well as more recently, in Diketopyrrolopyrrole based conductive polymers[51–54].

**Time evolution in the nanomechanical properties of C16-IDTBT device-grade thin films**. We now turn our attention to the evolution of the nanomechanical properties with time and the role of the residual solvent. We saw that exposing C16-IDTBT devices to ambient air led to trap healing, improved the electrical properties by increasing the hole carrier mobilities, and reduced the entropy per carrier. Regarding its nanomechanical properties, there is no difference between the evolution of organic films stored in ambient air and organic films stored in a nitrogen glovebox. Also, in contrast to C16-IDTBT's electronic property evolution, the polymer's nanomechanical properties stabilize on a longer time scale owing to a mechanism pinned not to oxygen diffusion, but to a gradual sweating-out of residual, low molecular weight solvent molecules from the organic polymer film's surface.

The amount of residual low molecular weight solvent compounds in the polymer film depends on the film's initial annealing conditions. In fabricating high mobility organic transistors from C16-IDTBT, the best electronic device characteristics are obtained when the solution-processed polymer thin film is annealed at 100 °C[55]. Annealing the film at higher temperatures has a demonstrated effect in worsening the device characteristics[55]. Since the film is solution processed from the solvent 1,2 dicholorobenzene (DCB) which has a boiling point of 180 °C, the thin film of C16-IDTBT continues to hold on to residual solvent molecules within it even after being annealed for up to an hour at 100 °C. The vapor pressure of 1,2 DCB at 25 °C is 0.00186 bar and the vapor pressure at 100 °C is 0.0847 bar as calculated from the Antoines equation. The low vapor pressure at room temperature suggests slow evaporation. Although it has been shown that solvent retention can temporarily enhance the charge transport properties of C16-IDTBT based transistors, similar to a molecular additive[11,12], the same solvent retention has an adverse effect on the nanomechanical properties of the C16-IDTBT thin films. This is on account of the film surface having a much higher stickiness, or high adhesion forces in the presence of the solvent. The presence of solvent also results in a viscoelastic response under load, which is manifested as a hysteresis between force curves measured on approach and retraction. This, in turn, renders determination of the elastic modulus from an elastic contact mechanics model, such as the Derjaguin Muller Toporov (DMT) model, questionable[56,57]. The film thus needs to be allowed to sweat out the residual solvent molecules from its surface over time, to correctly present the pristine nanomechanical properties of the polymer film (See Supplementary Information Sections 9 and 10).

Figure 4(a) and (b) show surface plots of the nanomechanical properties of a C16-IDTBT thin film processed under identical conditions as those used in the electrical and thermoelectric devices above, but which was left undisturbed under ambient conditions (and in the dark to avoid photo oxidative effects) for several weeks to allow the residual low molecular weight solvent molecules to evaporate from the surface under their own volatility. The microscale topography, stiffness, adhesion, and elastic modulus of the stabilized film were acquired using Park System's PinPoint Nanomechanical measurement mode on an NX20 AFM with a cantilever tip radius of 10 nm. The technique extracts the various nanomechanical properties using the captured force curves which map the interaction between the AFM tip and the sample surface on approach and on retraction[58–62]. The films have a smooth texture, with the average modulus quantified to be around 2 GPa. Over the scan of 5 $\mu$m $\times$ 5 $\mu$m as shown in Fig. 4(a), the modulus appears highly homogeneous confirming our earlier report[63]. Figure 4(b) plots the nanoscale topography, stiffness, adhesion, and modulus of the stabilized C16-IDTBT film but on a length scale of a few hundred square nanometers. On this length scale, significant texture

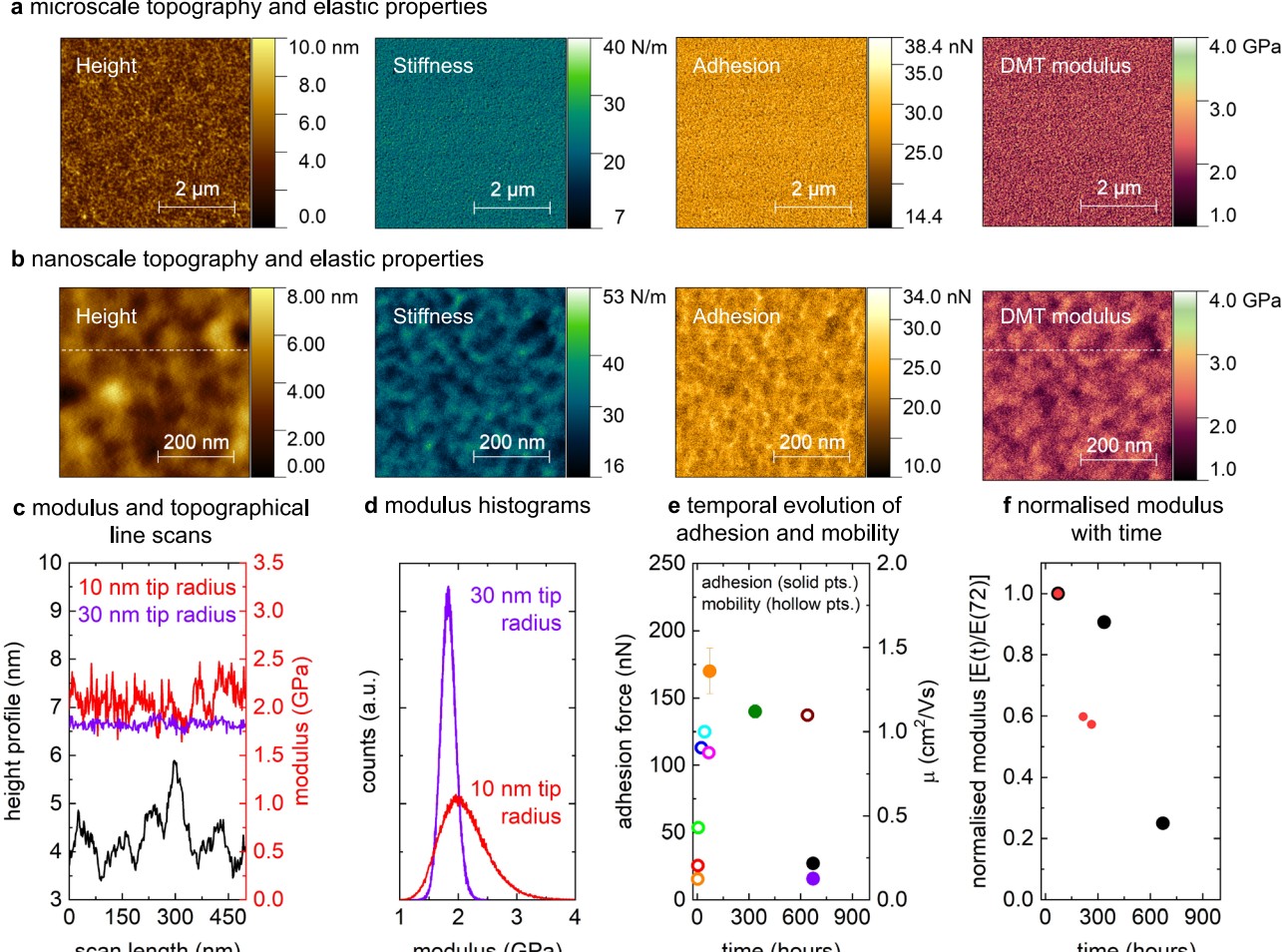

**Fig. 4 Nanoscale mechanical properties and nanomechanical homogenization with time. a** Topography and elastic properties (stiffness, adhesion, and modulus) in C16-IDTBT on the microscale. **b** Topography and elastic properties (stiffness, adhesion, and modulus) in C16-IDTBT on the nanoscale. The thin films were processed under identical conditions as those used in the electrical and thermoelectric devices but were left undisturbed under ambient conditions and in the dark for several weeks to allow the low molecular weight solvent molecules to evaporate from the surface under their own volatility. **c** Line scan of the modulus and topography along the dotted white lines shown in **(b)** measured using a tip radius of 10 nm. A scan of the modulus with a tip radius of 30 nm is shown in purple for comparison[63]. **d** Comparison of modulus histograms measured with two different tip radii; a 30 nm tip radius (larger than ordered film features) and a 10 nm tip radius (equivalent or less than ordered film features). **e** Comparison showing how the film adhesion changes far slower with time than the saturation mobility. **f** Normalized modulus measured on different C16-IDTBT samples using two different AFMs. The black dots correspond to the measurements on a Park Systems NX20 and the red dots correspond to measurements on a JPK NanoWizard 3 Atomic Force Microscope. The modulus reduction and homogenization that accompanies the reduction in adhesion force takes place on a time scale longer than the stabilization of the saturation mobility.

emerges in the stiffness, adhesion, and the modulus. The average modulus remains similar on these small scales. To understand whether the apparent texture in the nanomechanical modulus seen at high resolution arises from differences between the ordered and disordered regions of C16-IDTBT on the nanometer scale, Fig. 4(c) plots line scans of the topography and the modulus along the dotted white lines of Fig. 4(b). The topography registers a broader undulation over 100 nm. The modulus line scan remains uncorrelated with the topography which is an indication that the probed texture is not an artefact of the tip used in the measurements. In addition, the nanomechanical modulus probed using the sharper 10 nm tip shows significant features on the scale of 50 nm or less, whereas the same films probed with a 30 nm tip radius charted a far more uniform modulus[63]. As shown in Fig. 4(d), the histogram asymmetry presents a tail with higher modulus values in the enhanced resolution measurement (using a 10 nm tip) indicating potential crystallinity and pointing at differences between the ordered and disordered regions being

probed on the stabilized film. When the probe tip has a radius of 30 nm, the tip dimension is well above the fine structural features in the film, and hence only an average nanomechanical property is probed with a narrow symmetric spread in the measured nanomechanical response, again shown in Fig. 4(d). The texture in the nanoscale modulus of C16-IDTBT at high resolution is thus a true film feature, and the spread in values about the average seen in the modulus line scans and modulus histograms confirm the presence of both order and disorder within the thin film.

A C16-IDTBT film that has not been left to stabilize over time presents different nanomechanical properties to those discussed above. In general, we find that the adhesion force and the elastic modulus decrease with time. For instance, the first adhesion map was measured 3 days after fabrication of the film and showed a very high surface adhesion force of 170 nN, suggesting either capillary condensation of the solvent between tip and sample or bridging of mobile polymer chains to the tip. As discussed earlier

and as shown in Fig. 4(e), within the first 3 days, the electronic properties of the film already stabilize to show relatively ideal characteristics and high mobilities. The film's adhesion however continues to dynamically change over several weeks at a much more gradual pace, and in keeping with the out gassing of volatile organic solvent molecules from the film as shown in Fig. 4(e). After a month, the films achieve a nominal value in the adhesion of ~27 nN, which compares well with measurements carried out on fully stabilized thin films after several weeks[63]. Clearly, at this stage the solvent content has decreased to such low levels that capillary condensates do not form and bridging of polymer chains no longer occurs due to decreased polymer chain mobility at the surface. These latter adhesion data correlates with additional nanomechanical measurements (see Supplementary Information Section 9), where an adhesion force of similar values was measured after several weeks of sample manufacturing. The reduction in the surface adhesion on the nanoscale occurs on a slower time scale compared to the healing in its electronic properties. Figure 4(f) illustrates the relative change in average modulus, defined as $E(t)/E(72)$ where $E(72)$ is the average modulus measured initially after 72 h and $E(t)$ the modulus measured at later times. One set of data was measured simultaneously with the adhesion shown in Fig. 4(e), since both these properties are extracted from the same force curve within the PinPoint nanomechanical measurement on Park System's NX20 AFM. A subsequent set of data was determined on another sample using JPK's Quantitative Imaging mode. Considering that two different samples were analyzed using two different instruments and AFM modes, the agreement between the two data sets is satisfactory and unequivocally demonstrates a significant reduction in modulus with time. The modulus change that occurs over a month is a consequence of solvent sweating out from the film, a process that takes much longer than it takes for the hole mobility to achieve its benchmark threshold value of 1 cm$^2$/Vs. A dynamic process involving bubble formation on the film surface in the presence of additional solvent, or over time, was also captured (see Supplementary Information Section 9).

The combined nanoscale mapping of the topography, phase, adhesion, and modulus carried out in this work reinforce growing evidence that C16-IDTBT contains regions of high order, in addition to regions of disorder on the nanoscale. For the process of oxygen diffusion that improves the electrical properties as well as the process of gradual sweating out of solvent molecules that contributes to a dynamic improvement in the measurable nanomechanical properties, the amorphous regions of the film might play an important role. This is because percolation of oxygen and solvent molecules in and out of the film's matrix may be easier in the less ordered regions. This assumption is speculative and demands comprehensive investigation, the sort of which goes beyond this manuscript.

In this work, we spotlight the time evolution of the electrical, thermoelectric and the nanomechanical properties of an archetypical high mobility polymer semiconductor under ambient conditions. We utilize the established device processing protocol that guarantees high carrier mobilities, and tie together with it an understanding of its nanomechanical properties. The salient outcomes of our work that significantly further the organic electronics community's understanding of organic semiconductors are as follows. First, we show direct evidence that thin spin-coated polymer films of C16-IDTBT present nanoscale order with domain sizes in the range of ten to fifteen nanometers. Second, we demonstrate that the electrical and the thermoelectric properties of C16-IDTBT bear a parametric dependence on time with a new scaling law; a linear proportionality between the Seebeck coefficient and the saturation mobility. The evolution of the device characteristics toward ideality over time is mediated by oxygen diffusion and doping in ambient air, followed by microstructure re-arrangement upon solvent removal, leading to very small subthreshold swings within transistors, reduction in trap densities and consequently, increased fractions of carriers that participate in entropy transport. Third, we demonstrate that the nanoscale mechanical modulus in C16-IDTBT thin films shows texture when probed on the scale of the individual ordered domains. This drives one to conclude that the nanocrystalline regions are stiffer than the amorphous regions in the film. Fourth, we present evidence for the gradual sweating out of solvent molecules from the thin film on a time scale longer than that required for electronic mobility stabilization. This effect is observed through what we purport as surface modulus homogenization. Owing to the observed dynamics over a month, our study promotes the need for well documented conditions around which the multifunctional properties of organic polymer devices are being reported in the literature.

## Methods

**Fabrication of transistor and Seebeck devices and its measurement sequence.** The 20 micron-wide gold source and drain electrodes are micropatterned using optical lithography and fabricated using thermal evaporation. A resistive stripe heater is simultaneously patterned within 20 microns from one of the electrodes. The stripe heater permits a measurement of the Seebeck coefficient within the transistor channel. In the device, the 50 nm thick organic semiconductor C16-IDTBT is patterned to overlap only the two electrodes that bridge the 50-micron channel. This avoids electrical cross talk with the resistive stripe heater. The patterned organic layer is 200 microns long and 1 mm wide. The source and drain electrodes also act as on-chip resistive thermometers for the measurement of the temperature gradient across the channel of the organic transistor. A 500 nm thick dielectric layer of Cytop-M (CTL-809M) is solution-processed over the patterned organic semiconductor and over the whole substrate. A 25 nm thick, 1.5 mm wide, several millimeters long gold gate electrode is deposited over the dielectric layer and covers the patterned organic semiconductor underneath it. Within this single device, both the transistor characteristics as well as the Seebeck coefficient modulated using a gate voltage are measured in a carefully monitored time sequence (see Supplementary Information Section 4). During device fabrication, every step was carefully executed to limit ambient air exposure. Immediately after spin coating the dielectric, a several hour pump down in vacuum was carried out before the final gate evaporation. The device was then transferred into a low temperature probe station and its transistor transfer and output characteristics were measured under vacuum. This measurement was followed by a measurement of the gate-voltage modulated Seebeck coefficient in vacuum. These collective measurements constituted the first set of measurements summarized in Fig. 2 under the heading "after fabrication". The sample chamber of the cryostat was then vented to permit ambient air into the chamber. The device was left in air for a documented time interval. At the end of the time interval of ambient exposure, the sample chamber of the cryostat was pumped down to vacuum and the sequence of transistor and Seebeck measurements was performed once again. After each subsequent set of measurements in vacuum, the sample chamber was vented to expose the sample to ambient air. Detailed steps in the fabrication of thin films and microscale devices from C16-IDTBT are documented in the Supplementary Information. Detailed steps on the electronic and thermoelectric measurement sequence used in this work, as well as the nuances of device characterization and parameter extraction are also documented in the Supplementary Information.

**Pinpoint nanomechanical measurements and AFM based phase topography.** AFM measurements were performed in the ambient at 22 °C on Park Systems NX20 Atomic Force Microscope (Park Systems Co., Suwon, Korea). High resolution phase maps were acquired using a novel higher eigenmode approach [V. V. Korolkov et al., Nat. Commun. **10**, 1537 (2019)] with Multi75Al-G (Budget Sensors, Bulgaria) cantilevers excited at the 3rd eigenmode. Nanomechanical data was collected with PinPoint Nanomechanical mode [S. Kim et al., Nanomaterials **11**(6), 1593 (2021)] on the same instrument. PinPoint Nanomechanical mode allows simultaneous acquisition of high-resolution topographical maps and force-distance curves at every single pixel of an image. Further automated analysis of these force-distance curves provides with nanomechanical properties such as modulus, adhesion, deformation, stiffness, and energy dissipation. For all nanomechanical measurements we used an AC160TS cantilever (Olympus, Japan) with a nominal spring constant of 42 N/m. Each probe was calibrated prior to measurements to find the spring constant and optical sensitivity. The C16-IDTBT thin film sample investigated was "stabilized", i.e., it was left to sweat out solvent molecules from its surface for a few weeks before the measurement could be reliably done.

## Data availability

The data used in this study are presented in the text and the Supplementary Information. Additional data are available from the corresponding authors upon reasonable request. The Atomic Force Microscopy and Nanomechanics dataset used in Fig. 1 and Fig. 4, are available at Cambridge University's online repository: https://doi.org/10.17863/CAM.84546.

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

## Acknowledgements

D.V. acknowledges the Royal Society for funding in the form of a Royal Society University Research Fellowship (Royal Society Reference No. URF\R1\201590). D.V. is thankful to Prof. Henning Sirringhaus for having facilitated this work by providing a cleanroom for device fabrication, a nitrogen glovebox deposition facility to deposit high purity organic thin films, and a desert cryogenics probe station using which the electrical and thermoelectric properties of C16-IDTBT were investigated. D.V. is particularly grateful to Prof. Henning Sirringhaus for having supported his research career till the point of independence. D.V. thanks Prof. Iain McCulloch for having supplied the C16-IDTBT used in this work. D.S. acknowledges support from the Sensor CDT and the Engineering and Physical Sciences Research Council (Grant No. EP/L015889/1). This work has also received funding from the European Union's Horizon 2020 research and innovation programme under grant agreement No 964677. O.D.J. and M.W. were supported by the National Science Foundation under award 1810273.

## Author contributions

I.D. and V.V.K. contributed equally to this work. I.D. and V.V.K. performed the nanomechanical measurements on C16-IDTBT presented in this work. V.V.K. implemented the higher eigen mode imaging technique to demonstrate nanoscale order in C16-IDTBT thin films. D.S., L.J.S., and D.V. fabricated the organic polymer thin films used in this study and engaged in regular discussions on the interpretation of the experimental measurements. D.V. fabricated and measured the transistor characteristics and the gate voltage modulated Seebeck coefficients in C16-IDTBT based devices and uncovered the parametric relationship between the mobility and the Seebeck coefficient. O.D.J. and M.W. carried out the trap density of states analysis. V.L. and Y.O. performed the molecular dynamics simulations. H.-I.U. explained the vanishing kink effect in the C16-IDTBT organic transistors based on trap state passivation. D.V. and P.M.C. co-ordinated and supervised the experimental collaboration between KTH Stockholm and the University of Cambridge. D.V. and P.M.C. wrote the final paper together with inputs from all the authors.

## Competing interests

The authors declare no competing interests.
