## [Peer Review File · Nature Communications]

Dynamic self-stabilisation in the electronic and nanomechanical properties of an organic polymer semiconductor having nanoscale orderREVIEWER COMMENTS

Reviewer #1 (Remarks to the Author):

The manuscript titled "Dynamic self-stabilisation in the electronic and nanomechanical properties of a near-amorphous organic polymer semiconductor" by the group of Claesson and Venkateshvaran offer a new insight into the time evolution of the physical properties of an archetypical high mobility organic polymer semiconductor under ambient conditions. The manuscript presents comprehensive and novel work in the field of polymer science, by combining electrical/thermoelectric characterisation with nanomechanical properties. This makes the manuscript timely, as both the areas of polymer electronic as well as nanomechanics are very topical at the moment.

In their work, the group of Claesson and Venkateshvaran show the importance of oxygen/solvent molecules on the time evolution of both electrical/thermoelectric and nanomechanical properties in thin spin-coated polymer films of C16-IDTBT. They present the four important results ;

1stly, in C16-IDTBT films, they show direct evidence of nanoscale order with domain sizes in the range of ten to fifteen nanometers.

2ndly, they measured the time evolution of the electrical and the thermoelectric properties of C16-IDTBT, and found a linear proportionality between the saturation mobility and Seebeck coefficient, which might be explained by oxygen diffusion and doping in ambient air.

3rdly, the nanoscale mechanical properties in C16-IDTBT thin films were investigated, and they conclude that the nanocrystalline regions are stiffer than the amorphous regions.

4thly, they also measured the time evolution of nanomechanical properties and present evidence for the gradual sweating out of solvent molecules from the thin film. Importantly, a time scale of nanomechanical property is longer than that required for electronic properties.

All these findings are very interesting and, in particular, the linking between electrical/thermoelectric properties with nanomechanical properties is scientifically new and important aspect. The reasoning seems sound and logical. The choice of the materials (C16-IDTBT) suits their narrative.

I have a few minor comments that I would like to see addressed before publication of this manuscript.

1. As I commented above, I believe that one of the most important findings is the relationship between the time evolution of electrical/thermoelectric properties and nanomechanical properties. As authors concluded, the origin of time evolution in electrical/thermoelectric properties and nanomechanical properties are oxygen and solvent molecules, respectively. As the results, a time scale of nanomechanical property is longer than that required for electronic properties. Although this difference is well presented in manuscript, to make their relationship clear, please try to make new plot which shows the relationship between them. For example, the plot between saturated mobility (or carrier density dependence of Seebeck coefficient) and adhesion force (or modulus) is one possibility. I would recommend the authors to consider the impressive plot which explain well the time scale difference between electrical/thermoelectric properties and nanomechanical properties.

2. In Figures 3g and 3h, as authors commented, $dS/d(\log n)$ is approaching to the slope of Jonker plot (-198 $\mu\text{V}/\text{K}/\text{dec}$). This is very reasonable and agrees with the previous wonderful results, reported by the group of Venkateshvaran (doi:10.1038/nature13854). However, this slope completely different from the recent results of highly doped donor-acceptor copolymer films (doi: 10.1103/PhysRevResearch.2.043330). I know many things are different among them. Even though, the authors should have a few lines to discuss the observed difference.

3. My final point is still under unclear. I don't have a firm answer myself but I raise this to encourage discussion. For both the oxygen diffusion and the gradual sweating out of solvent molecules, the

amorphous regions of film might play important roles and the near-amorphous structure of C16-IDTBT might be the true origin of the observed time evolution. If authors agree with this point, although the authors already commented this point slightly, for the sake of the organic polymer science community, I would recommend that authors add a couple of lines for the importance of amorphous regions. This will definitely add a new, much needed perspective to the field of polymer science.

I have no other comments. The nanomechanical properties of the paper is very scientifically sound. Most of my comments focus on the electrical/thermoelectric properties of the paper. After the suggested changes to the manuscript are made, I would like to make a recommendation to the editor that this paper should be published in Nature Communication.

Reviewer #2 (Remarks to the Author):

The authors report a detailed study of the behavior of IDTBT transistors with aging and exposure to ambient. The TFTs show the unusual response of a steepening sub threshold slope despite an apparent doping process.

I have a number of technical comments as follows:

(1) The authors discuss the role of exposure to air as a major factor in the modification in performance. The authors should specify more detail about how the experiments were done, i.e. in complete darkness?, in a laboratory environment or in a controlled environment of air. The authors point out that oxygen is a likely caused based on literature, but there are also contrary reports where the nature of the ambient was explored, e.g.

Chabinyk, Michael L., Robert A. Street, and John E. Northrup. "Effects of molecular oxygen and ozone on polythiophene-based thin-film transistors." *Applied physics letters* 90, no. 12 (2007): 123508.

Given the complexity of identifying the origin of defect states, the authors may want to comment on other possibilities given the presence of residual chlorinated solvents as well.

(2) The TFT characteristics in Figure 2c change in an unusual way for a doping process. The onset voltage appears to increase and then decrease over time as the on-current increases. This behavior would not be expected for simple doping where one would likely expect doping in the body of the semiconductor to push the onset voltage more positive over time. Could this change be a sign of multiple processes occurring with aging?

(3) I found the discuss of the AFM-based mechanical analysis confusing as written, e.g. what is the "DMT model"?; what is being reported as the "modulus", is it only an elastic response rather than a loss component? There is also text in the SI that suggests some of these sorts of measurements have been carried out previously. What is new here?

(4) I do not understand the sentence in the conclusion that "we build a strong case for a class of material that can be classified as near amorphous" given that in the manuscript the authors corroborate substantial local ordering of the polymer.

Reviewer #1 (Remarks to the Author):

The manuscript titled "Dynamic self-stabilisation in the electronic and nanomechanical properties of a near-amorphous organic polymer semiconductor" by the group of Claesson and Venkateshvaran offer a new insight into the time evolution of the physical properties of an archetypical high mobility organic polymer semiconductor under ambient conditions. The manuscript presents comprehensive and novel work in the field of polymer science, by combining electrical/thermoelectric characterisation with nanomechanical properties. This makes the manuscript timely, as both the areas of polymer electronic as well as nanomechanics are very topical at the moment.

In their work, the group of Claesson and Venkateshvaran show the importance of oxygen/solvent molecules on the time evolution of both electrical/thermoelectric and nanomechanical properties in thin spin-coated polymer films of C16-IDTBT. They present the four important results;

1stly, in C16-IDTBT films, they show direct evidence of nanoscale order with domain sizes in the range of ten to fifteen nanometres.

2ndly, they measured the time evolution of the electrical and the thermoelectric properties of C16-IDTBT, and found a linear proportionality between the saturation mobility and Seebeck coefficient, which might be explained by oxygen diffusion and doping in ambient air.

3rdly, the nanoscale mechanical properties in C16-IDTBT thin films were investigated, and they conclude that the nanocrystalline regions are stiffer than the amorphous regions.

4thly, they also measured the time evolution of nanomechanical properties and present evidence for the gradual sweating out of solvent molecules from the thin film. Importantly, a time scale of nanomechanical property is longer than that required for electronic properties.

All these founding are very interesting and, in particularly, the linking between electrical/thermoelectric properties with nanomechanical properties is scientifically new and important aspect. The reasoning seems sound and logical. The choice of the materials (C16-IDTBT) suits their narrative.

I have a few minor comments that I would like to see addressed before publication of this manuscript.

We genuinely thank the reviewer for very succinctly summarising the major outcomes of our work, for having acknowledged its timeliness, and for supporting the new research direction of organic semiconductor nanomechanics that we have brought together with organic transistors and organic thermoelectrics in this manuscript. The simultaneous dynamic evolution of the electrical, thermoelectric and the nanomechanical properties of functional organic polymers is not present in existing literature, and we have addressed that void.

Below, we attempt to answer the various concerns raised by the reviewer in as much detail as possible. In addition to the revisions suggested by the reviewer, all of which I have taken and incorporated into the revised manuscript, I have also showcased new simulations from a seldomly used transport model for thermoelectrics based on polaronic hopping at high carrier densities. I think this modelling effort was necessary to adequately support my response to the reviewer's second question.

My response to the reviewers is documented in blue below.

1. As I commented above, I believe that one of the most important findings is the relationship between the time evolution of electrical/thermoelectric properties and nanomechanical properties. As authors concluded, the origin of time evolution in electrical/thermoelectric properties and nanomechanical properties are oxygen and solvent molecules, respectively. As the results, a time scale of nanomechanical property is longer than that required for electronic properties. Although this difference is well presented in manuscript, to make their relationship clear, please try to make new plot which shows the relationship between them. For example, the plot between saturated mobility (or carrier density dependence of Seebeck coefficient) and adhesion force (or modulus) is one possibility. I would recommend the authors to consider the impressive plot which explain well the time scale difference between electrical/thermoelectric properties and nanomechanical properties.

The reviewer has raised a very good point here in presenting the temporal correlation between the electrical and the nanomechanical properties better. In acknowledgement of this fantastic suggestion, I have now re-done Fig 4 panels (d), (e), and (f). The new figure together with the updated caption is displayed below.

In Fig 4 (d), we show a comparison of the modulus histogram in a stabilized C16-IDTBT film measured with two different AFM probe tip radii. One tip radius (30 nm) is larger than the ordered features of the film and the second tip radius (10 nm) is on the order or smaller than the ordered features.

In a new Fig 4 (e), we have taken the direct suggestion of the reviewer and have plotted against time both the saturation mobility (as hollow points) on the right-hand y-axis and the adhesion force (as solid points) on the left-hand y-axis. This plot shows clearly how the mobility stabilization has taken place well before the adhesion achieves its stable value of approximately 27 nN. In a new Fig 4 (f), we plot the normalised modulus of the film as a function of time. On this plot, we have also included new measurements of the modulus that we took using another Atomic Force Microscope from JPK Instruments. This new data, shown in red, confirms that the measured modulus depends sensitively on the condition of the film and its residual solvent content.

(a) microscale topography and elastic properties

(b) nanoscale topography and elastic properties

(c) modulus and topographical line scans

(d) modulus histograms

(e) temporal evolution of adhesion with mobility

(f) normalised modulus with time

Figure 4 Nanoscale mechanical properties and nanomechanical homogenization with time. (a) Topography and elastic properties (stiffness, adhesion, and modulus) in C16-IDTBT on the microscale. (b) Topography and elastic properties (stiffness, adhesion, and modulus) in C16-IDTBT on the nanoscale. The thin films were processed under identical conditions as those used in the electrical and thermoelectric devices but were left undisturbed under ambient conditions and in the dark for several weeks to allow the low molecular weight solvent molecules to evaporate from the surface under their own volatility. (c) Line scan of the modulus and topography along the dotted white lines shown in Fig. 4 (b) measured using a tip radius of 10 nm. A scan of the modulus with a tip radius of 30 nm from [63] is shown in purple for comparison. (d) Comparison of modulus histograms measured with two different tip radii; a 30 nm tip radius (larger than ordered film features) and a 10 nm tip radius (equivalent or less than ordered film features). (e) Comparison with time showing how the film adhesion changes far slower than the saturation mobility. (e) Normalized modulus measured on different C16-IDTBT samples using

two different AFMs. The black dots correspond to the measurements on a Park Systems NX20 and the red dots correspond to measurements on a JPK NanoWizard 3 Atomic Force Microscope. The modulus reduction and homogenization that accompanies the reduction in adhesion force takes place on a time scale longer than the stabilization of the saturation mobility.

In addition to these changes, we have also updated the Supplementary Information SI Section 9 on the nanomechanical properties and have included confirmatory measurements of the modulus that we measured using a third sensitive AFM based technique called Intermodulation AFM. All these confirmatory measurements prove that the stable value of the modulus of C16-IDTBT is 2 GPa and not larger.

Our paper now spotlights three different modes of nanomechanical measurement in total, namely Pinpoint Nanomechanical mode, Quantitative Imaging mode, and Intermodulation AFM mode, and all these techniques point to a stable value in the measured modulus that evolves only upon releasing the residual solvent.

All text changes associated with the above discussion have been highlighted in the main manuscript.

2. In Figures 3g and 3h, as authors commented, $dS/d(\log n)$ is approaching to the slope of Jonker plot (-198 $\mu\text{V}/\text{K}/\text{dec}$). This is very reasonable and agrees with the previous wonderful results, reported by the group of Venkateshvaran (doi:10.1038/nature13854). However, this slope completely different from the recent results of highly doped donor-acceptor copolymer films (doi: 10.1103/PhysRevResearch.2.043330). I know many things are different among them. Even though, the authors should have a few lines to discuss the observed difference.

The reviewer has raised a very important point regarding the measured slopes of the Seebeck coefficient as a function of carrier density. This has indeed been an overlooked point in experiments carried out on gated organic thermoelectrics. I have thought about this in the last few years myself and will attempt to clarify the issue in this letter.

Below, I start by laying down selected observations in the experiments, both of ours, and of other research groups. I shall then briefly discuss the Jonker model together with its applicability and its failings. Within this context, I will highlight the findings of the Jonker analysis when applied to small polaron systems such as oxides and organic semiconductors in different transport regimes. Finally, I shall build a coherent summary that is consistent with the current findings in the field of organic thermoelectrics.

The observations are listed sequentially as follows.

Their interrelation will become clear as the discussion evolves.

(a) **Comparison of Cytop-gated IDTBT with ion-gel electrolyte gated DPPs.**

I've taken a few representative data points of the Seebeck coefficient vs carrier density in the various DPPs from the work highlighted by the reviewer, namely K. Kanahashi *et al.*, Physical Review Research **2**, 043330 (2020), and have plotted this data together with the Seebeck data of IDTBT presented in our own manuscript.

This plot is shown below in Fig R1.

Figure R1 Seebeck versus conductivity in different transport regimes

With respect to Fig R1, the following three features are evident.

1, the slope of the Seebeck coefficient vs conductivity in IDTBT is larger than $-\frac{k_B}{e}\ln(10) \approx -198 \mu\text{V/K}$ in magnitude and the slopes of the DPPs on the same plot are a fraction of $-\frac{k_B}{e}\ln(10)$. These slopes are recalculated here based on the visually extracted data from the paper by K. Kanahashi ... T. Takenobu *et al.*, but they agree well with the said publication.

2, the magnitude of the Seebeck coefficient of IDTBT is several times larger than the DPPs.

3, in the very same range of conductivity, i.e., around $\sigma \approx 1 \text{ S/cm}$, both IDTBT and the DPPs behave very differently in slope and in magnitude.

In addition to the insights from the above figure, another important piece of information comes from the work of Broch and Venkateshvaran [K. Broch, D. Venkateshvaran *et al.*, *Advanced Electronic Materials* **3**, 1700225 (2017)], where the Seebeck coefficients in various DPP based polymers were measured within OFET architectures. In fact, some of the DPP polymers measured in our prior work overlap with Kanahashi-san and Takenobu-sensei's Physical Review Research paper, yet the Seebeck coefficients we measured were larger in magnitude and had a Seebeck slope steeper than $-\frac{k_B}{e}\ln(10)$. For example, in our previous work we reported the Seebeck coefficient of DPPT-TT to be $324 \mu\text{V/K}$ with a slope of $-277 \mu\text{V/K}$ per decade in the hole regime.

These joint observations point to the fact that the very same polymer can show two different trends depending on whether they are gated using a solid-state layer such as Cytop and PMMA, or whether they are gated/doped using an ion gel electrolyte layer such as [DEME][TFSI]/P(VDF-HFP) and its cousins.

(b) **$S - \log(\sigma)$ in the non-degenerate regime.**

In our own measurements on IDTBT, we performed Seebeck measurements in the regime of a gated-thin film transistor where the carrier density n is between 10^{18} and 10^{19} cm^{-3} and is much smaller than the total number of available states N . In other words, $n \ll N$. Under this condition, transport is non-degenerate. In addition, assuming transport in narrow bands (as is the case for the low-torsion polymer IDTBT), the Seebeck coefficient $S \approx \frac{k_B}{e}\ln\left(\frac{N}{n}\right)$. This justifies the Seebeck slope being $-\frac{k_B}{e}\ln(10)$, in the absence of traps and on a linear $S - \log(n)$ plot.

Also, since the mobility in IDTBT is gate voltage invariant as seen in D. Venkateshvaran *et al.*, Nature **515**, 384 (2014), the ideal slope of $S - \log(\sigma)$ should be $-\frac{k_B}{e} \ln(10)$ in the trap-free case (more on this later), and the slope should be larger in the presence of a small amount of disorder.

The latter is validated through the slope on Fig R1 of the $S - \log(\sigma)$ plot above, and in the main manuscript.

(c) **Implicit assumption of Jonker Analysis.**

The Jonker analysis, first shown in G. H. Jonker, Philips Res. Rep. **23**, 131 (1968), derives quantitative relationships between transport parameters in semiconductors of either p-type, n-type, or mixed conduction by plotting 'pear-shaped' curves between the Seebeck coefficient and the conductivity. Several insights into the conduction parameters can be got using 'Jonker-pears', such as the DOS-mobility product and the intrinsic bandgap of the semiconductor.

G. H. Jonker clearly mentions in his work that the analysis is applicable to non-degenerate semiconductors where Boltzmann statistics can be applied. In addition, he unequivocally states in his paper that an uncertainty in the construction of $S - \log(\sigma)$ can arise for higher doping concentrations. For this reason, in his paper, Jonker generates the plot of $S - \log(\sigma)$ in PbS with carrier densities that go up to only 10^{19} cm^{-3} .

(d) **Deviation of Jonker slopes in SWCNT networks.**

With the above description of Jonker's original work from 1968, it comes as no surprise that there can indeed be circumstances that cause deviations from the ideal slope value of $-\frac{k_B}{e} \ln(10)$ in a plot of $S - \log(\sigma)$. Such deviations, showing a slope lower than $-\frac{k_B}{e} \ln(10)$ in magnitude, have indeed been seen in several materials.

For example, in single walled carbon nanotube networks such as those shown in M. Statz *et al.*, ACS Nano **14**, 15552 (2020), a deviation is seen at large carrier densities greater than 10^{19} cm^{-3} . At these large carrier densities, configurational entropy does not primarily govern the Seebeck coefficient and Heikes formula is violated. In the large carrier density regime, the Seebeck slope $< -\frac{k_B}{e} \ln(10)$ in magnitude. In the words of the first author of this paper, "the reduced carrier concentration dependence of the Seebeck coefficient is a manifestation of Boltzmann statistics becoming insufficient due to the Fermi level (E_F) approaching the state distribution ($n \ll N$ does not hold). In this regime a narrow-band description is inappropriate, and the actual DoS of the networks needs to be considered."

(e) **Deviation of Jonker slopes in multifunctional oxides.**

Another class of materials where the Seebeck slope $< \frac{k_B}{e} \ln(10)$ has been demonstrated is in the multifunctional oxides, for example, in amorphous InGaZnO shown in J. C. Felizco *et al.*, Applied Surface Science **527**, 146791 (2020). In this work, the slopes on the $S - \log(\sigma)$ plot are between $54 \mu\text{V/K}$ and $70 \mu\text{V/K}$, i.e., much smaller than the expected ideal slope of $200 \mu\text{V/K}$. The authors of that work ascribe this behaviour to hydrogenation of InGaZnO which renders its conduction metallic. A further report on the same material is shown in AIP Advances **5**, 097209 (2015).

In a second example, Koji Watari-san and co-workers from AIST in Nagoya have shown in Y. Kinemuchi *et al.*, J. Mater. Res. **22**(7) 1942 (2007), that the ideal slope of $S - \log(\sigma)$ is also reduced in Al-doped ZnO.

A third example in the oxides where the Seebeck slope $< \frac{k_B}{e} \ln(10)$ is shown in J. Nell *et al.*, Journal of Solid State Chemistry **82**, 247 (1989). Here, the Jonker analysis is used to distinguish small polaron hopping from band-type semi-conductivity. The authors of this work studied several oxide based systems, from spinel ferrites and non-stoichiometric oxides to highly doped

systems such as $\text{LaCrO}_3\text{:Sr}$ and $\text{Mn}_3\text{O}_4\text{:Fe}$ and discuss the strengths and limitations of the Jonker analysis.

(f) **Evidence of small polaron conduction in oxides and polymers using Jonker analysis**

Figure R2 (a) below shows plots of the Seebeck coefficient vs the log of conductivity. It compares the ideal Jonker plot (shown as a dashed line) for a band semiconductor in the non-degenerate transport regime, with both p-type and n-type polaronic systems. This figure (a) is extracted from J. Nell *et al.*, Journal of Solid State Chemistry **82**, 247 (1989).

As the Seebeck coefficient gets lower and lower, accompanied by larger and larger conductivities, the Seebeck coefficient begins to show a curvature with a steeper trend on this plot. A similar trend in the Seebeck coefficient with the conductivity that deviates from the ideal Jonker lines was demonstrated in the comprehensive research work conducted in the group of Prof Takenobu at Nagoya University. Fig (b) and Fig (c) below show two plots extracted from the paper H. Tanaka *et al.*, Sci. Adv. 6 : eaay8065 (2020) on electrolyte gated PBTTT, and the paper K. Kanahashi *et al.*, Physical Review Research **2**, 043330 (2020) on ion-gel gated electrolyte DDP polymers, respectively. Very clearly, a steeper trend in the Seebeck coefficient is seen at the highest achievable conductivities (accentuated here by the log-log scale of course). The papers postulated the existence of an insulator to metal like transition when the Seebeck deviates from the empirical trend $S \propto \sigma^{-1/4}$ to the trend $S \propto \sigma^{-1}$.

This said, such a change in trend is also consistent with the manifestation of the small polaron behaviour shown in Fig R2 (a) below. In addition, as seen in Fig R2 (b) and Fig R2 (c) below, the power factor $S^2\sigma$ registers a maxima when plotted as a function of conductivity, deviating from the empirical trend of $S^2\sigma \propto \sigma^{1/2}$ shown in doped polymers such as PEDOT and PANI in B. Russ *et al.*, Nature Reviews Materials **1**,16050 (2016).

In Fig R2 (d), I have taken the polaronic model of Fig R2 (a) and plotted it on a log-log scale here for the Seebeck coefficient versus normalised conductivity. I have also computed and plotted the power factor vs normalised conductivity. The polaronic model that I used in this plot takes the form $\sigma/\sigma_{\text{max}} = 0.5^2 \times q/(1 + q)^2$ where $q = 2 \times \exp(-Se/k_B)$ for a p-type polaronic system. As can be seen, the simulated plots in Fig R2 (d) come very close to resembling the experimental data of Fig R2 (c) in all respects.

Figure R2 Seebeck and Power Factor versus conductivity, beyond Jonker and beyond Heikes

The maxima in the power factor may be systematically also explained within an alternative modified Jonker formalism that gives rise to reduced slopes in the $S - \log(\sigma)$.

This alternative explanation for the same is below.

In some semiconducting systems exhibiting polaronic transport, such as Al-doped ZnO, the mobility is observed to increase proportional to increases in the conductivity. In other words, $\mu \propto \alpha \sigma$ where α is a constant of proportionality. When this observation is seen, the slope of $S - \log(\sigma)$ goes from being $-\frac{k_B}{e} \ln(10)$ to being $-\frac{k_B}{e} \ln(10) \times (1 - n\alpha)$. Such a 'reduced Seebeck slope' was indeed measured in Al-doped ZnO in the manuscript, Y. Kinemuchi *et al.*, J. Mater. Res. **22**(7) 1942 (2007).

Beyond the oxide hopping conductors, a trend like $\mu \propto \alpha \sigma$ is seen in various F4TCNQ and TFSI doped PBTBT polymers shown in S. Watanabe *et al.*, Physical Review B **100**, 241201(R) (2019). I have extracted data from this paper and plotted it in Fig R3 (a) below. In addition, the same trend was also measured in the ion-gel [EMIM][TFSI] gated organic semiconductor C8-DNBDT-NW as shown in the manuscript: N. Kasuya *et al.* Nature Materials **20**, 1401 (2021). I have extracted data from this paper too and plotted it below in Fig R3 (b).

Figure R3 Mobility vs conductivity relationships

My region of interest in the above plots of Fig R3 is in the high conductivity region of several 10s S/cm. The mobility values shown above are the measured Hall mobilities.

As can be seen in these above two plots, the proportionality coefficient $\alpha = 0.00413 \text{ cm}^3/(\text{S-Vs})$ for F4TCNQ doped PBTBT and $\alpha = 0.026 \text{ cm}^3/(\text{S-Vs})$ for ion-gel gated C8-DNBDT.

Using the modified Jonker slope reference above of $-\frac{k_B}{e} \ln(10) \times (1 - ne\alpha)$ together with these values of α , and with carrier densities $n \sim 2 \times 10^{20} \text{ cm}^{-3}$ as is the case for both the doped and the ion-gel gated samples, one finds that the F4TCNQ doped sample will not have much of a reduction of the expected slope since $-\frac{k_B}{e} \ln(10) \times (1 - 2 \times 10^{20} \times 1.6 \times 10^{-19} \times 0.00413) = -0.87 \frac{k_B}{e} \ln(10)$, whereas in the ion-gel gated organic semiconductor C8-DNBDT, $-\frac{k_B}{e} \ln(10) \times (1 - 2 \times 10^{20} \times 1.6 \times 10^{-19} \times 0.026) = -0.17 \frac{k_B}{e} \ln(10)$.

$ne\alpha$ is dimensionless since $\text{cm}^{-3} \times \text{C} \times \text{cm}^3/(\text{S-Vs})$ all cancel out.

In other words, the reduced Seebeck slope in the high conductivity regime is justified by a modification of the Jonker slope to the form of $-\frac{k_B}{e} \ln(10) \times (1 - ne\alpha)$, and is very much α -dependent rather than dependent on carrier concentration alone.

What α was in the paper by K. Kanahashi *et al.*, Physical Review Research **2**, 043330 (2020) is difficult to extract since the Hall mobilities were not published in the same paper, but my suspicion is that it would behave like the ion-gel gated C8-DNBDT, and thus justify low Seebeck slopes.

When $\mu \propto \alpha\sigma$, the Jonker formula for the Seebeck coefficient which is originally $S = -\frac{k_B}{e} \{-\ln\sigma + \ln[e\mu_0 N_c \exp(A)]\}$, needs to be modified. It becomes $S = -\frac{k_B}{e} \{-\ln\sigma + \ln(\mu_0 + \alpha\sigma) + \ln[eN_c \exp(A)]\}$. All parameters hold their usual meanings in the context of semiconductor thermoelectrics in these equations. In simpler terms, the Seebeck coefficient can now be written as $S = \frac{k_B}{e} (1 - K) \times (\ln\sigma - C)$ where K and C are empirical constants. This equation indicates that the Seebeck coefficient measured in such a regime is also reduced in magnitude compared to the original predictions of Jonker's formula for S stated above.

Using this equation for the Seebeck coefficient, a maximum exists in the power factor since taking $\frac{\partial S^2\sigma}{\partial \sigma} = 0$ yields and optimum/maximum σ_{optimum} in the $S^2\sigma$ vs σ plot.

I sum up using in the words of Nell in J. Nell *et al.*, Journal of Solid State Chemistry **82**, 247 (1989); “Curvature at large carrier concentrations in the $S - \log(\sigma)$ plot, i.e., for low absolute values of thermopower, can be taken as diagnostic for small polaron conduction. The sign of thermopower at the nose of the curve indicates the sign of the polaron, whether n-type or p-type.”

What all the above details taken together equates to is simple.

When a doped semiconductor system registers a steep change in trend in the Seebeck vs conductivity plot at high carrier density, this sharp reduction can be explained by the existence of polaronic conduction. Polarons are known to exist in organic semiconductors on account of their weak van der Waals bonding and polarisation character. In such polaronic systems, the relationship between mobility and conductivity (i.e., $\mu \propto \alpha\sigma$) and the magnitude of the proportionality constant is crucial.

In the presence of a sizable α , ($> 0.01 \text{ cm}^3/(\text{S-Vs})$ for example) and for carrier densities $> 10^{19} \text{ cm}^{-3}$, the Seebeck slope on the $S - \log(\sigma)$ curve can be smaller in magnitude than $-\frac{k_B}{e} \ln(10)$, and the new ‘reduced Seebeck slope’ is $\frac{\partial S}{\partial \log \sigma} = -\frac{k_B}{e} \ln 10 \times (1 - ne\alpha)$. First evidence for such behaviour in a polaronic system was shown in J. Mater. Res. **22(7)** 1942 (2007). In the regime of these observations, the power factor of the system also registers a maxima.

A maximum in the power factor, a reduced Seebeck slope, and a diminished magnitude of the Seebeck coefficient were all seen in the published works of Prof Takenobu’s group from Nagoya shown previously in this letter. A maximum in the power factor and a reduced Seebeck slope are indirect signatures of polarons in the organic material, in my own opinion.

The conductivities are intrinsically limited by the polaronic diffusivity that exists in these materials, and the magnitude of the conductivity pre-factor in the variable range hopping model for these systems can be a very telling parameter.

Lastly, Prof David Emin, in his papers C. Wood and D. Emin, Physical Review B **29(8)**, 4582 (1984) and D. Emin, Phys. Rev. Lett. **35** (13), 882 (1975), postulates that when conduction occurs in an energy band of small polaron states within amorphous semiconductors, the Seebeck coefficient develops an additional term that is directly proportional to temperature.

In other words, the current conundrum in the organic thermoelectrics literature where at high charge carrier densities, the Seebeck shows a direct proportionality with temperature, accompanied by conductivities that continue to be hopping like, may concomitantly indicate evidence of small polaron transport (more on this in the next two sections).

(g) Seebeck coefficient versus Temperature trend in doped organic polymers

The understanding of electronic transport in doped organic polymers is complicated by experimental evidence that does not follow a single transport mechanism. The conductivity can be either thermally activated or show variable range hopping, but the Seebeck coefficient exhibits a ‘metallic-like’ temperature dependence. A recent work that conclusively showed this was by the group of Takeya and Watanabe, namely, S. Watanabe *et al.*, Physical Review B **100**, 241201(R) (2019). I have reproduced the plot below in Fig R4. Clearly, the Seebeck coefficient is linear in temperature, often said to obey the Mott formula, but corresponding conductivity does not show a metallic-like behaviour.

Figure R4 Conductivity and Seebeck coefficient versus temperature

There continues to be only scattered reports that provide conclusive evidence on what is fundamentally going on in these doped polymeric systems with respect to Seebeck and conductivity. More experimental evidence will be necessary to build a solid case that goes beyond simplistic empirical trends in the $S - \sigma$ relationships such as those proposed by Kang and Snyder.

(h) **Small polaron Seebeck coefficient and its direct proportionality to Temperature**

Without going into too much further detail, I wish to say here that the work of D. Emin on small polarons might provide a fertile ground that unifies all the seeming contradictory results on carrier hopping and transport in organic semiconductors. D. Emin believes that in the presence of structural disorder such as in amorphous semiconductors, or where hopping occurs between inequivalent sites, an energy band of small polaron states occurs resulting in an additional term which is approximately linear in temperature such that $S' = S + \beta T$. S' is the Seebeck coefficient, S is the standard configurational entropy term and β is a constant of proportionality. In particular, the overall Seebeck coefficient S' measured for such small polarons is much larger (~ 10 s to few $100 \mu\text{V/K}$) than the Seebeck expected for a degenerate semiconductor or a metal with a well-defined Fermi sea ($\sim 10 \mu\text{V/K}$). Linear trends in the Seebeck coefficient with temperature in small polaron systems are indeed seen in materials such as Silicon Carbide as shown in C. Wood and D. Emin, *Physical Review B* **29**(8), 4582 (1984). A theoretical formalism for the same is shown in D. Emin, *Phys. Rev. Lett.* **35**(13), 882 (1975). In Prof Emin's own words "For weakly degenerate small-polaron hopping among a broad distribution of inequivalent sites the Seebeck coefficient is reasonably close to the observed form, $S' = S + \beta T$."

The configurational entropy term S reduces with higher carrier concentration of course.

(i) **Summary**

When considering charge and thermoelectric transport within doped organic semiconductors within the comprehensive framework of small polarons, the apparent contradictions in the various trends can be put to rest. One such contradiction is the hopping mechanism in conductivity with a Seebeck trend that is proportional to T and seemingly 'metallic-like'. The work of Prof Takenobu in *Science Advances* (2020) and in *Physical Review Research* (2020) show beautifully that the Seebeck coefficient vs conductivity and the power factor vs conductivity in highly conductive organic semiconductors displays trends which I have shown in this letter as being governed by polaronic transport theories. Very similar trends have been seen in polaronic systems such as the amorphous conductive oxides. The slope of the Seebeck coefficient versus log conductivity in such 'weakly degenerate' systems having high doping levels that do not show thermally activated hopping can indeed be less $< \frac{k_B}{e} \ln(10)$ in magnitude if the quantity $ne\alpha$ starts to become appreciable.

Some of the exposition above goes well beyond the scope of the current manuscript. However, aided by the clarity they afford, I have now introduced the following lines in the manuscript to acknowledge that there are indeed recent reports in chemically doped and ion-gel electrolyte gated organic semiconductors which show slopes of $< -\frac{k_B}{e} \ln(10)$.

The following lines have now been included in the manuscript

The ideal slope of the Seebeck coefficient on a Jonker plot, i.e., $S - \log(n)$ or $S - \log(\sigma)$, is $-\frac{k_B}{e} \ln(10)$ within the non-degenerate transport regime where the carrier density n is well below the total number of available states N . This is typically the case in our Cytop-gated organic field-effect transistors where the carrier density is around $10^{18} - 10^{19} \text{ cm}^{-3}$. In systems such as ion-gel electrolyte gated organic transistors, the carrier density is typically an order of magnitude or larger than it is in Cytop-gated organic field-effect transistors. At such high carrier densities which approach a 'near-degenerate' transport regime, the slope of the Seebeck coefficient is known to deviate and be shallower than $-\frac{k_B}{e} \ln(10)$. Within a transport model for polaronic hopping that has routinely been used to explain transport in multifunctional conductive oxide-based semiconductors, the magnitude of reduction in the Seebeck slope is known to depend sensitively on the dimensionless quantity $ne\alpha$. Here, α is the proportionality constant between the carrier mobility and the conductivity in a high conductivity regime, e is the electronic charge and n is the carrier density. In high conductivity polaronic systems, this reduced Seebeck slope becomes $-\frac{k_B}{e} \ln(10) \times (1 - ne\alpha)$, and has been observed in several multifunctional conductive oxide-based semiconductors, as well as more recently, in Diketopyrrolopyrrole (DPP) based conductive polymers. [51], [52], [53], [54]

The following reference have been included in this context

- [51] K. Kanahashi, Y.-Y. Noh, W.-T. Park, H. Yang, H. Ohta, H. Tanaka and T. Takenobu, "Charge and thermoelectric transport mechanism in donor-acceptor copolymer films," *Physical Review Research*, vol. 2, p. 043330, 2020.
- [52] J. Nell, B. J. Wood, S. E. Dorris and T. O. Mason, "Jonker-type analysis of small polaron conductors," *Journal of Solid State Chemistry*, vol. 82, p. 247, 1989.
- [53] Y. Kinemuchi, C. Ito, H. Kaga, T. Aoki and K. Watari, "Thermoelectricity of Al-doped ZnO at different carrier concentrations," *J. Mater. Res.*, vol. 22, no. 7, p. 1942, 2007.
- [54] J. C. Felizco, M. Uenuma, Y. Ishikawa and Y. Uraoka, "Optimizing the thermoelectric performance of InGaZnO thin films depending on crystallinity via hydrogen incorporation," *Applied Surface Science*, vol. 527, p. 146791, 2020.

All text changes associated with the above discussion have been highlighted in the main manuscript.

3. My final point is still under unclear. I don't have a firm answer myself but I raise this to encourage discussion. For both the oxygen diffusion and the gradual sweating out of solvent molecules, the amorphous regions of film might play important roles and the near-amorphous structure of C16-IDTBT might be the true origin of the observed time evolution. If authors agree with this point, although the authors already commented this point slightly, for the sake of the organic polymer science community, I would recommend that authors add a couple of lines for the importance of amorphous regions. This will definitely add a new, much needed perspective to the field of polymer science.

This is indeed another very valid point that the reviewer has raised.

I believe a conclusive picture that differentiates the role of the amorphous regions from the crystalline regions in the temporal stabilisation of the electrical properties and the nanomechanical properties is difficult to gain without doing measurements on site-specific locations of the film that have a confirmed presence of either disordered or ordered regions. It is also difficult to say whether the nanoscopic regions of order were present right from the beginning, at the time of film fabrication and when the

residual solvent content was high... or whether the regions of order were 'grown' during the process of adiabatic solvent out-sweating.

We now show in a new Supplementary Information Section 6 that upon gradual sweating out of the residual solvent, there may be an adiabatic structural reorganisation within the film. A hint of this can be seen in the extracted electronic density of states after month, highlighted in SI Section 6. In addition, The measurements of the modulus and adhesion force over time show a gradual reduction in their histogram's full width at half maximum. This is additional evidence that there is a continuous nanoscale homogenisation taking place within the film as a function of time (see error bars in Fig 4 (e) which reduce with time). Such a continuous change within the film makes it difficult to put a finger on exactly how the amorphous and the nanoscale ordered regions behave since, as speculatively mentioned earlier, the nanoscale order might be grown/seeded over time as the residual solvent is very slowly released.

Since the picture of what is going on in the film can quickly become very complex by considering multiple speculative contributions, we have chosen to focus on the averaged measurables of adhesion and modulus in addition to the steady state nanomechanical maps of the film. This has been the implicit message of Fig. 4. All the nanoscale order and disordered regions and surface maps that we show in our manuscript were measured on stable films with low residual solvent content.

The ordered regions on the film's surface immediately after fabrication were impossible to measure to be honest. This is because the residual solvent content was high, the adhesion was high, and the sample was too sticky to measure any surface order, despite the sensitivity of our higher eigen mode imaging technique.

With the above clarification, we have now added the following lines to the manuscript.

For the process of oxygen diffusion that improves the electrical properties as well as the process of gradual sweating out of solvent molecules that contributes to a dynamic improvement in the measurable nanomechanical properties, the amorphous regions of the film might play an important role. This is because percolation of oxygen and solvent molecules in and out of the film's matrix may be easier in the less ordered regions. This assumption is speculative and demands comprehensive investigation, the sort of which goes beyond this manuscript.

All text changes associated with the above discussion have been highlighted in the main manuscript.

I have no other comments. The nanomechanical properties of the paper is very scientifically sound. Most of my comments focus on the electrical/thermoelectric properties of the paper. After the suggested changes to the manuscript are made, I would like to make a recommendation to the editor that this paper should be published in Nature Communication.

All the authors on our paper thank the reviewer once again for the positive and supportive assessment of our combined research effort.

Reviewer #2 (Remarks to the Author):

The authors report a detailed study of the behavior of IDTBT transistors with aging and exposure to ambient. The TFTs show the unusual response of a steepening sub threshold slope despite an apparent doping process.

I have a number of technical comments as follows:

(1) The authors discuss the role of exposure to air as a major factor in the modification in performance. The authors should specify more detail about how the experiments were done, i.e. in complete darkness?, in a laboratory environment or in a controlled environment of air. The authors point out that

oxygen is a likely caused based on literature, but there are also contrary reports where the nature of the ambient was explored, e.g.

Chabinyk, Michael L., Robert A. Street, and John E. Northrup. "Effects of molecular oxygen and ozone on polythiophene-based thin-film transistors." *Applied physics letters* 90, no. 12 (2007): 123508.

Given the complexity of identifying the origin of defect states, the authors may want to comment on other possibilities given the presence of residual chlorinated solvents as well.

The reviewer has raised a valuable question. I would like to expand upon our measurements below:

1. Both the transistor characterisation and the Seebeck measurements were performed in a reconfigured desert cryogenics probe station with optical access. The device had a few wires connected to it via an external high-vacuum feed-through, but also made use of the four available micromanipulator probe arms fitted with the BeCu 10-micron probe tips for electrical measurements. The device was designed in a fashion to have the electrical contact pads several millimetres away from the active area of the device.
2. The additional wires were necessary to make contact because the device has 9 contact pads; one gate electrode and four for each resistance thermometer which also doubled back as source and drain electrodes. The device sat on a temperature-controlled metal chuck throughout the measurements. To accommodate for the extra wiring, the inner aluminium chamber that sits within the vacuum chamber was not covered with a UV filter. The outer vacuum chamber contained the optical access window necessary for repositioning the probe tips during measurement under vacuum. What this meant is that the device was exposed to light during and intermittently when the lab lights were on, even if not continuously. A lamp light was shone in the optical access when the electrical probes needed to recontact the device. Because of the way the experiments were setup and conducted, we must account for the presence of light during the healing/aging process.
3. The detailed protocol on how the measurements were conducted constitute parts of our Supplementary Information and the Methods section, but very briefly and to be specific to the line of questioning here, I wish to point out the following. The sample was pumped down to vacuum (around 10^{-5} mbar) in the desert cryogenics low temperature probe station where the transistor transfer curves and the Seebeck measurements were conducted. Measurements in vacuum were necessary to avoid hysteresis. The chamber was then vented to allow ambient air into it. A documented amount of time was allowed to pass after which the chamber was pumped down once again, and the measurement cycle was repeated. For the measurements we took after a month, the device was removed from the cryostat and reinstalled at the time of final measurement. During this time the sample was left in ambient atmosphere and exposed to laboratory lighting.
4. Just as in the reference pointed out by the reviewer, multiple contributions can influence the changes seen in the device and they most likely occur simultaneously, but on different time scales. I detail the various contributors below:
 - (a) There is the influence of ambient oxygen in the presence of light that we have tried to use here as justification, based on previous reports.
 - (b) There is the influence of small amounts of ozone which can play a similar role as oxygen, I understand from the reference mentioned by the reviewer.
 - (c) There could be a potential influence of water incorporation from atmospheric humidity that could enter the device over time and degrade its performance (something we don't observe, most likely because Cytop provides a good barrier for water).
 - (d) There is the influence of solvent sweating on account of residual DCB leaving the device over time (which we have tried to measure using the adhesion channel in nanomechanics).

- (e) There could be structural reorganisation within the film upon solvent sweating. It is difficult to measure the nanoscopic order in fresh films with solvent because the surface of the film can be quite sticky initially. We can only measure the surface nanoscale order after many weeks. For this reason, it has been difficult to claim either way that the IDTBT polymer film contains nanoscale order from the outset, or whether there is a slow ordered growth over time (the latter is a speculation that we now make based on additional trap-density of states simulations in a new Supplementary Information Section 6)
- (f) There could be an influence of other trace gasses in the atmosphere whose direct impact is difficult to quantify.
- (g) In the spirit of identifying all potential causes for the presence of trap states, I would even like to point out that the photolithographic patterning process used to fabricate the Seebeck devices (explained in detail in the Supplementary Information), might also be a cause of trace amounts of initially degraded performance, although such degradation has been strongly contested in the literature. The photolithography process was documented earlier by a former member of the Optoelectronics group in Cambridge, J. F. Chang *et al.*, *Adv. Funct. Mater.* **20**, 2825–2832 (2010), and showed no sign of such anticipated degradation. The understanding is that throughout the patterning process, the organic is protected by a relatively thick bilayer of Cytop/photoresist, and this acts like a protective layer for the C16-IDTBT layer.
- (h) The presence of any residual chlorinated solvents in the film can form water solvent azeotropes as already mentioned in our manuscript through the referenced work of M. Nikolka *et al.*, *Adv. Mater.* **30**, 1801874 (2018). This if anything, is meant to remove the adverse effects of water incorporation in the device initially.

It is precisely for the potential multitude of reasons that can affect an organic transistors performance initially, that we attempted to look at the temporal evolution of the transistor characteristics of devices fabricated from established procedures with documented/guaranteed high mobility. We consider all the influences mentioned above as intrinsic to the nature of the device when measured for the first time after fabrication under as clean and controlled conditions as possible within our laboratory.

5. We also know two additional pieces of important information that we have already mentioned in our manuscript, namely, (a) An IDTBT device fabricated at the same time and left in a nitrogen glovebox will not improve its transistor characteristics unless exposed to ambient air, and (b) A thin film of IDTBT deposited with the same device parameters does not need ambient air exposure to normalise its nanomechanical properties. These nanomechanical properties are normalised when stored in nitrogen, or when exposed to ambient. It didn't seem to matter.
6. The two findings in point 5 above were what caused us to consider ambient air as a source of device healing, because any trace solvent incorporation will be released from the film irrespective of storage conditions, unless strongly chemically bound.

We have attempted to clarify the various sources of contamination above and the reasons why we focused on ambient oxygen as a source of electronic trap healing. We have also included a new supplementary information section, SI Section 6, that investigates the trap density of states extracted directly from our transistor measurements. These simulations are explained within the context of the reviewer's next question.

Within the context of the above explanation and of the pertinent reference on the role of oxygen and ozone on polythiophenes that the reviewer has pointed us to, we have incorporated the following text and reference in the main manuscript.

Doping induced by ambient exposure or controlled exposure to ppm quantities of ozone are known to cause a shift in the threshold voltage in organic transistors towards more positive values as reported in

some polythiophenes. [43] Such a doping process is normally accompanied by degraded transistor performance, evidenced by high off-currents and by sub-threshold slopes that get shallower upon exposure. In our work, the observation that the threshold voltage initially becomes more positive does indeed indicate doping, but the observation that it eventually reverses direction within a month accompanying a steepening in the subthreshold slope is an indication that, at the very least, one other accompanying process takes place. We speculate that this accompanying process might be related to a structural reorganization within the film as the residual solvent contained within it sweats out over time. Evidence of the same is presented in the context of the nanomechanical property measurements later in this paper.

[43] M. L. Chabinyk, R. A. Street and J. E. Northrup, "Effects of molecular oxygen and ozone on polythiophene-based thin-film transistors," *Applied Physics Letters*, vol. 90, no. 12, p. 123508, 2007.

In addition, we have now incorporated a discussion in SI Section 4 surrounding details of how the measurements were performed. This should clarify in depth, the role of ambient light, as well as the other influences that might contribute to what we see in the time evolution of the electrical measurements.

All text changes associated with the above discussion have been highlighted in the main manuscript.

(2) The TFT characteristics in Figure 2c change in an unusual way for a doping process. The onset voltage appears to increase and then decrease over time as the on-current increases. This behavior would not be expected for simple doping where one would likely expect doping in the body of the semiconductor to push the onset voltage more positive over time. Could this change be a sign of multiple processes occurring with aging?

This is a question related to our previous discussion. Yes, it is indeed true that there may be multiple processes occurring with aging and not just a single doping process. To clarify this in greater depth, we teamed-up with the group of Prof Oana Jurchescu who was able to elucidate the multiple processes occurring during ageing by calculating the trap density of states (t-DOS) of C16-IDTBT. These t-DOS curves were extracted directly from the measured transfer characteristics of our device that we present in Fig. 2 of the main paper.

The spectral analysis of trap density of states now constitutes an exclusive section in the Supplementary Information, namely SI Section 6.

In Fig. S6 (a) of the new SI Section 6, also shown in this letter below, we include a representative curve from three different states of the device as it heals upon air exposure. In Fig. S6 (b) also shown below, we include the complete set of curves along with their corresponding air exposure times (offset for clarity).

The trap-DOS follows an exponential distribution typical for polycrystalline and amorphous organic semiconductors, where the electronic states tailing into the bandgap are the result of structural defects and/or impurities present in the film. The peak found in the curves immediately after fabrication confirms that discrete electronic states are present in the bandgap. These states are eliminated upon extended exposure to ambient air, as evidenced by the fact that the peak is gradually suppressed. Additionally, the increase in the density of states very close to the top of the valence band (shallow traps) supports the hypothesis that oxygen doping occurs during the first several hours of exposure, *i.e.*, in the first regime (initial 10 hours). Later, the overall density of trap states is reduced, most likely due to microstructural changes taking place in the film upon solvent removal, as discussed in detail in the main manuscript.

Additional details on how these trap-DOS simulations were calculated are summarised in SI Section 6. We have also included a new reference in the main paper within this context, namely,

[18] H. F. Iqbal, M. Waldrip, H. Chen, I. McCulloch and O. D. Jurchescu, "Elucidating the Role of Water-Related Traps in the Operation of Polymer Field-Effect Transistors," *Advanced Electronic Materials*, vol. 7, p. 2100393, 2021.

Fig S6 Trap DOS spectra evaluated at different times during device operation. (a) A subset of 3 curves, each corresponding to one state of the device and highlighting the dynamic processes occurring in the film. (b) Complete dataset offset for clarity. All curves are plotted as a function of energy from the valence band edge.

Prof Jurchescu and her PhD student Matthew Waldrip have been added as co-authors of the current manuscript for their contribution.

All text changes associated with the above discussion have been highlighted in the main manuscript. The following lines have been included.

In addition, the initial increase in the onset voltage, which is the signature of doping, followed by a decrease over time [Fig. 2 (c)], suggests that two different processes occur during device aging. Indeed, the density of trap states (t-DOS) spectra confirm the presence of two regimes (see Supplementary Information Section 6). [18], [27]

An exact elucidation of the nature of the trap states seen in our electronic device measurements goes beyond the current scope of this manuscript. We have nevertheless attempted to prove here that there is indeed more than just one simple contribution to the trap healing that is going on in the density of states over time.

(3) I found the discuss of the AFM-based mechanical analysis confusing as written, e.g. what is the "DMT model"?; what is being reported as the "modulus", is it only an elastic response rather than a loss component? There is also text in the SI that suggests some of these sorts of measurements have been carried out previously. What is new here?

We thank the reviewer for pointing out this insufficiency. We have now re-written significant parts of the discussion on the nanomechanical properties presented in this work. In this connection, the first reference to the DMT model now reads "the Derjaguin Muller Toporov (DMT) model", and we have included the following two landmark references on this frequently used model.

[56] B. V. DERJAGUIN, V. M. MULLER and Y. P. TOPOROV, "Effect of Contact Deformations on the Adhesion of Particles," *Journal of Colloid and Interface Science*, vol. 53, no. 2, p. 314, 1975.

[57] D. S. GRIERSON, E. E. FLATER and R. W. CARPICK, "Accounting for the JKR–DMT transition in adhesion and friction measurements with atomic force microscopy," *J. Adhesion Sci. Technol.*, vol. 9, no. 3-5, p. 291, 2005.

The DMT model is a model used in surface mechanics that nowadays is commonly utilized to extract the modulus of a film from a measured force-distance curve between an AFM tip and the sample

surface. One such example of a force-distance curve is shown in Fig S10 of Supplementary Information SI Section 9. The DMT model is fit to the short range repulsive and adhesive regime of this curve recorded during surface indentation. The DMT model is ubiquitously used for materials that have a dominant elastic contribution. In the presence of a strong viscoelastic contribution, the approach and retract region of the force-distance curve will show significant hysteresis. In the SI Section 9, we have attempted to justify in a few words as to why the DMT model can indeed be used in our measurements. When the viscoelastic contribution is large, in other words, when the sample contains a large amount of solvent, the DMT model will yield unreliable results. It is for this reason that we have only shown results in Fig 4 (a), (b), (c) and (d) on films that were left to stabilise over several weeks so that the contribution from the surface solvent to the modulus is negligible. In a new Fig 4 (e), the high adhesion force in samples measured within days from fabrication is shown. This result does not come from the linear region of the force curve, but from the curvature shown at negative (attractive) forces. In Fig 4 (f), we have plotted the normalised modulus as a new plot in the revised manuscript. The data points are normalised to the first data point measured after three days (72 hours) from fabrication. The reason we plot the normalised modulus and not the absolute value of the modulus is precisely because the absolute values cannot be trusted in this regime where residual solvent in the films yield a strong viscoelastic contribution. The value of the modulus that can be trusted from the DMT model is the value of 2 GPa that we report on stabilized films. It is not the value of around 8 GPa that is extracted from the films immediately upon fabrication.

As a side note, part of the intent of our manuscript is to showcase the subtle effects in the nanomechanical property estimation of organic semiconductor films processed in a way to achieve high mobility. As the field of AFM-based polymer nanomechanics expands in reach, we would like to encourage accurate estimation and reporting of the nanomechanical properties. What we would like to avoid is a situation that plagued organic transistors for many years where the mobilities were extracted from non-optimum device characteristics to yield ambitious and incorrect values of the mobility. We attempt to highlight in our paper the need for better control and understanding of nanomechanical measurements as well as the measured values themselves.

SI Section 9 has now been re-written completely to prove that the modulus of C16-IDTBT, extracted from the DMT model which considers only an elastic response, is indeed appropriate in our films that have been let to sweat out any residual solvent. We have now incorporated into this SI Section 9 additional measurements from a Quantitative Imaging mode as well as from a very powerful new development called Intermodulation AFM that separates elastic and viscous contributions during tip-sample interactions. Using both these additional techniques, we show that the modulus of C16-IDTBT is indeed 2 GPa. We also confirm once again that high resolution nanomechanical texture is evident using the Intermodulation AFM mode. The modulus texture is on the expected order of the ordered regions in the film.

SI Section 10 shows nanomechanical measurements on C16-IDTBT films contaminated with PDMS-like plasticizers. None of these results have been published earlier in any shape or form. The previous reference that we made to the “work of D. Simatos (2022)” was a reference to Dr. D. Simatos’ unpublished PhD thesis in which he measured the modulus of C16-IDTBT with plasticizers within a week from fabrication only. What Dr Simatos had not measured for his PhD thesis was the same sample several weeks after allowing the residual solvent to evaporate. This is the new add-on investigation that has been shown in Fig. S12 (e, f, g, h).

In other words, what is new in this SI section that is of direct relevance to our manuscript is a comparison of the time evolution of C16-IDTBT nanomechanical properties in the presence of PDMS-like plasticizers. None of this has been demonstrated elsewhere prior to our current manuscript. The section was incorporated into the SI to demonstrate the solvent sweating-out effect which we dwell on in the main paper. We show that it is predominantly seen in C16-IDTBT and not as much from the PDMS-like plasticizer. It is a result that showed that in the presence of the solvent, the DMT model yields an ‘apparent modulus’ value for the C16-IDTBT host of over 6 GPa owing to viscoelastic effects. After several weeks however, it is shown that the DMT modulus of C16-IDTBT achieves its nominal value of 2 GPa which is a purely elastic contribution.

The measurements of the modulus in SI Section 10 were done with a PeakForce Quantitative Nanomechanical Mode (QNM) to quantify the modulus. This is a fourth technique in addition to the Pinpoint Nanomechanical Mode, the QI mode and the Intermodulation AFM that were used in various

other parts of the main paper and the supplementary information. What we see from all these measurements, is that the value for the Young's modulus of C16-IDTBT in its stable form and after the removal of the residual solvent is 2 GPa.

All told, the high resolution nanomechanical measurements on C16-IDTBT films in our manuscript that demonstrate texture depending on the ordered and the disordered regions in the film, is a world's first. This remains one of the dominant themes of our current manuscript and has never been seen elsewhere prior to this submission to Nature Communications.

(4) I do not understand the sentence in the conclusion that "we build a strong case for a class of material that can be classified as near amorphous" given that in the manuscript the authors corroborate substantial local ordering of the polymer.

The reviewer has raised a pertinent point. The reason we used the phrase 'near-amorphous' in the first version of the manuscript was because it is a common phrase used in the community in connection with IDTBT-based reports. Examples of usage of the phrase "near-amorphous" were seen in recent papers by Zhenan Bao's group (*Adv. Funct. Mater.* 2019, 29, 1905340), Christine Luscombe's group (*Macromolecules* 2020, 53, 17, 7511–7518), as well as a few other noteworthy research groups.

In light of the nanoscale ordered features shown within this manuscript, I have now chosen to remove all references to the phrase 'near-amorphous'. Instead, I have used phrases such as 'nanoscale order' or 'disordered regions' and so forth, in its place. I have removed the sentence that was pointed out by the reviewer above. Lastly, the title of the manuscript has been changed to reflect this. The title of the manuscript now reads, "Dynamic self-stabilisation in the electronic and nanomechanical properties of an organic polymer semiconductor having nanoscale order".

Everything said, I wish to end this letter by profusely thanking both Reviewer #1 and Reviewer #2 for having taken the time to review our work. The questions and concerns raised by the reviewers have inspired careful thought, which we have attempted to address using additional work in the form of new trap density of states simulations, a new polaronic model for thermoelectric transport, and additional nanomechanical measurements using the Quantitative Imaging mode as well as using high-contrast intermodulation AFM. We have also tidied up the manuscript overall, based on the general inputs of both the reviewers. Lastly, we can confirm that none of the work contained in this manuscript or in its accompanying supplementary information has been made public in any shape or form.

As the corresponding author of this paper, I have made particular effort to ensure that our transatlantic team meets the high standards of comprehensiveness and reproducibility demanded of high-quality scientific research papers in general.

To this end, I sincerely hope that both the reviewers as well as the editor at Nature Communications find the major revisions which we have painstakingly carried out to meet their expectations.

All authors look forward with optimism to a favourable decision on our re-submitted manuscript.

With all best wishes,

Dr Deepak Venkateshvaran
Royal Society University Research Fellow, Cavendish Laboratory, University of Cambridge
Director of Studies in Physics, Selwyn College, University of Cambridge

REVIEWERS' COMMENTS

Reviewer #1 (Remarks to the Author):

In response to my comments, I strongly recommend the publication of this paper in Nature Communications, as it has been thoroughly revised by the authors.

Reviewer #2 (Remarks to the Author):

the authors have made a very comprehensive response to the comments from the reviewers. The manuscript is publishable in current form.

Reviewer #1 (Remarks to the Author):

In response to my comments, I strongly recommend the publication of this paper in Nature Communications, as it has been thoroughly revised by the authors.

Reviewer #2 (Remarks to the Author):

the authors have made a very comprehensive response to the comments from the reviewers. The manuscript is publishable in current form.

The authors thank both Reviewer #1 and Reviewer #2 for having taken the time to carefully assess our work, and for the suggestions on its overall improvement.

We also thank the editor at Nature Communications for having efficiently handled our manuscript and for guiding it from submission through peer-review.